# SUPERMODULAR RANK: SET FUNCTION DECOMPOSITION AND OPTIMIZATION

## ABSTRACT

We define the supermodular rank for set functions. This is the smallest number of terms needed to decompose it into a sum of supermodular functions. The supermodular summands are defined with respect to different partial orders. We characterize the maximum possible value of the supermodular rank and describe the functions with fixed supermodular rank. We analogously define the submodular rank. We use submodular decompositions to optimize set functions. Given a bound on the submodular rank of a set function, we formulate an algorithm that splits an optimization problem into submodular subproblems. We show that this method improves the approximation ratio guarantees of several algorithms for monotone set function maximization and ratio of set functions minimization, at a computation overhead that depends on the submodular rank.

*Keywords:* supermodular cone, imset inequality, set function optimization, greedy algorithm, approximation ratio

## 1 INTRODUCTION

We study the optimization of set functions – functions that are defined over families of subsets. The optimization of set functions is encountered in image segmentation (Boykov and Kolmogorov (2004)), clustering (Narasimhan et al. (2005)), feature selection (Song et al. (2012)), and data subset selection (Wei et al. (2015)). Brute force optimization is often not viable since the individual function evaluations may be expensive and the search space has exponential size. Therefore, one commonly relies on optimization heuristics that work for functions with particular structure. A classic function structure is supermodularity, which for a function on a lattice[1] requires that $f(x) + f(y) \leq f(x \wedge y) + f(x \vee y)$. A function is submodular if its negative is supermodular. Submodularity can be interpreted as a "diminishing returns" property. Submodularity and supermodularity can be used to obtain guarantees for greedy optimization of set functions, in a similar way as convexity and concavity are used to obtain guarantees for gradient descent/ascent. Well-known examples of results using supermodularity are the results of Nemhauser et al. (1978) for greedy maximization of a monotone submodular set function subject to cardinality constraints and those of Călinescu et al. (2011) for arbitrary matroid constraints. Refinements of these results have been obtained (Conforti and Cornuéjols (1984); Sviridenko et al. (2017); Filmus and Ward (2012)).

As many set functions of interest are not submodular or supermodular, relaxations have been considered, such as the submodularity ratio (Das and Kempe (2011)), generalized curvature (Conforti and Cornuéjols (1984); Bian et al. (2017); Buchbinder et al. (2014); Gatmiry and Gomez-Rodriguez (2018)), weak submodularity (Chen et al. (2018); Halabi and Jegelka (2020)), $\epsilon$-submodularity (Krause et al. (2008)), submodularity over subsets (Du et al. (2008)), or bounds by submodular functions (Horel and Singer (2016)). We will discuss some of these works in more detail in Section 4.

We propose a new approach to define the complexity of a set function. Specifically, we define the *supermodular rank* of a set function. We consider set functions that are real-valued with domain $2^{[n]} \cong \{0, 1\}^n$. Using a refinement of our notion of rank, we present a new algorithm for set function optimization via submodular decompositions. Our definition of complexity of set functions

---

[1]I.e., a poset where any two elements $x, y$ have a greatest lower bound $x \wedge y$ and a least upper bound $x \vee y$.

lets us take guarantees for low complexity functions and apply them to higher complexity functions.

**Main contributions.**

- We introduce the notion of supermodular rank for set functions (Definition 13). The functions of supermodular rank at most $r$ comprise a union of Minkowski sums of at most $r$ supermodular cones. We characterize the facets of these sums (Theorem 12) and find the maximum supermodular rank (Theorems 14) and maximum elementary supermodular rank (Theorem 19).
- We describe a procedure to compute low supermodular rank approximations of functions via existing methods for highly constrained convex optimization (Section 3).
- We show that the supermodular rank decomposition provides a grading of set functions that is useful for obtaining optimization guarantees. We propose the R-SPLIT and R-SPLIT RATIO algorithms for monotone set function and the ratio of set functions optimization (Algorithms 2 and 6), which can trade-off between computational cost and accuracy, with theoretical guarantees (Theorems 30, 72 and 73). The case for the ratio of set functions optimization is presented in Appendix F.3. These improve on previous guarantees for greedy algorithms based on approximate submodularity (Tables 2 and 3). We also provide a lower bound for the complexity of optimizing an elementary submodular rank-$(r + 1)$ function (Theorem 33).
- Experiments illustrate that our methods are applicable in diverse settings and can significantly improve the quality of the solutions obtained upon optimization (Section 5).[2]

**Other related work.** Supermodular functions are defined by linear inequalities. Hence, they comprise a polyhedral cone called the supermodular cone. These inequalities are called *imset inequalities* in the study of conditional independence structures and graphical models (Studený (2010)). They impose non-negative dependence on conditional probabilities. The supermodular cone has been a subject of intensive study (Matúš (1999); Studený (2001)), especially the characterization of its extreme rays (Kashimura et al. (2011); Studený (2016)), which in general remains an open problem. Supermodular inequalities appear in semi-algebraic description of probabilistic graphical models with latent variables, such as mixtures of product distributions (Allman et al. (2015)). Seigal and Montúfar (2018) suggested that restricted Boltzmann machines could be described using Minkowski sums of supermodular cones (Appendix I). We obtain the inequalities defining Minkowski sums of supermodular cones, which could be of interest in the description of latent variable graphical models.

## 2 SUPERMODULAR CONES

In this section we introduce our settings and describe basic properties with proofs in Appendix B.

**Definition 1.** Let $X$ be a set with a partial order such that for any $x, y \in X$, there is a greatest lower bound $x \wedge y$ and a least upper bound $x \vee y$ (this makes $X$ a lattice). A function $f \colon X \to \mathbb{R}$ is *supermodular* if, for all $x, y \in X$,

$$f(x) + f(y) \leq f(x \wedge y) + f(x \vee y). \tag{1}$$

The function $f$ is *submodular* (resp. *modular*) if $\leq$ in (1) is replaced by $\geq$ (resp. $=$).

**Example 2.** Let $X$ be the poset of all subsets of a set $S$ ordered by inclusion. Then a function $f$ is supermodular is for all $A, B \subset S$, we have that

$$f(A) + f(B) \leq f(A \cap B) + f(A \cup B).$$

Here $f(A) = |A|$ is a modular function. If $g$ is concave, then $f(A) = g(|A|)$ is submodular and if $h$ is convex then $f(A) = h(|A|)$ is supermodular.

**Definition 3.** We fix $X = \{0, 1\}^n$ and consider a tuple of linear orders $\pi = (\pi_1, \ldots, \pi_n)$ on $\{0, 1\}$. Our partial order is the product of the linear orders: for any $x, y \in X$, we have $x \leq_\pi y$ if and only if $x_i \leq_{\pi_i} y_i$ for all $i \in [n]$. For each $i$, there are two possible choices of $\pi_i$, the identity $0 \leq_{\pi_i} 1$, and the transposition $1 \leq_{\pi_i} 0$. A function that is supermodular with respect to $\pi$ is called $\pi$-*supermodular*.

---

[2]Computer code for our algorithms and experiments is provided in [anonymous GitHub repo].

**Example 4.** Consider $n = 2$, then $X = \{(0,0), (1,0), (0,1), (1,1)\}$. Then, suppose we have the standard linear orders, that is, $\pi_1$ and $\pi_2$ are the identity. Then we have that $(0,0)$ is the smallest element, $(1,1)$ is the biggest, and $(1,0)$ and $(0,1)$ are intermediate incomparable elements.

Now suppose $\pi_1$ is a transposition. This implies that $1 \leq_{\pi_1} 0$ for the first coordinate. Thus, we have that $(1,0)$ is the smallest, $(0,1)$ is the biggest, and $(0,0)$ and $(1,1)$ are the intermediate incomparable elements. Hence with this latice, we have that $f$ is supermodular if and only if

$$f((0,0)) + f((1,1)) \leq f((1,0)) + f((0,1)).$$

For fixed $X$, the condition of $\pi$-supermodularity is defined by requiring that certain homogeneous linear inequalities hold. Hence the set of $\pi$-supermodular functions on $X$ is a convex polyhedral cone. We denote this cone by $\mathcal{L}_\pi \subseteq \mathbb{R}^X$. Two product orders generate the same supermodular cone if and only if one is the total reversion of the other, and hence there are $\frac{1}{2} \prod_i |X_i|!$ distinct cones $\mathcal{L}_\pi$, see Allman et al. (2015). The cone of $\pi$-*submodular* functions on $X$ is $-\mathcal{L}_\pi$.

In the case $X = \{0,1\}^n$, the description of the supermodular cone in Equation 1 involves $\binom{2^n}{2}$ linear inequalities, one for each pair $x, y \in X$. However, the cone can be described using just $\binom{n}{2} 2^{n-2}$ facet-defining inequalities (Kuipers et al. (2010)). These are the elementary imset inequalities (Studený (2010)). They compare $f$ on elements of $X$ that take the same value on all but two coordinates. Identifying binary vectors of length $n$ with their support sets in $[n]$, the elementary imset inequalities are

$$f(z \cup \{i\}) + f(z \cup \{j\}) \leq f(z) + f(z \cup \{i,j\}), \tag{2}$$

where $i, j \in [n]$, $i \neq j$ and $z \subseteq [n] \setminus \{i,j\}$.

We partition the $\binom{n}{2} 2^{n-2}$ elementary imset inequalities into $\binom{n}{2}$ sets of $2^{n-2}$ inequalities, as follows.

**Definition 5.** For fixed $i, j \in [n]$, $i \neq j$, we collect the inequalities in Equation 2 for all $z \subseteq [n] \setminus \{i,j\}$ into matrix notation as $A^{(ij)} f \geq 0$. We call $A^{(ij)} \in \mathbb{R}^{2^{n-2} \times 2^n}$ the $(ij)$ *elementary imset matrix*.

We give examples to illustrate the elementary imset inequality matrices from Definition 5.

**Example 6** (Elementary imset inequalities). Given $f \colon \{0,1\}^n \to \mathbb{R}$, the elementary imset inequalities are

$$f_{\ldots 0 \ldots 1 \ldots} + f_{\ldots 1 \ldots 0 \ldots} \leq f_{\ldots 0 \ldots 0 \ldots} + f_{\ldots 1 \ldots 1 \ldots}, \tag{3}$$

where $f_{\ldots 0 \ldots 1 \ldots} := f(\cdots 0 \cdots 1 \cdots)$ and an index $(\cdots a \cdots b \cdots)$ has varying entries at two positions $i$ and $j$. Fixing $i$ and $j$, one has $2^{n-2}$ inequalities in equation 4. For example, if $i = 1$ and $j = 2$ then one obtains two inequalities:

$$f_{010} + f_{100} \leq f_{000} + f_{110}$$
$$f_{011} + f_{101} \leq f_{001} + f_{111}.$$

**Example 7** (Three-bit elementary imset inequality matrix). For $n = 3$, $A^{(12)}$ is the $2 \times 8$ matrix

$$A^{(12)} = \begin{matrix} & 000 & 001 & 010 & 011 & 100 & 101 & 110 & 111 \\ 0 & \begin{pmatrix} 1 & & -1 & & -1 & & 1 & \\ & 1 & & -1 & & -1 & & 1 \end{pmatrix} \\ 1 & \end{matrix}.$$

Each row of $A^{(ij)}$ has two entries equal to $1$ and two entries equal to $-1$.

The elementary imset characterization extends to $\pi$-supermodular cones $\mathcal{L}_\pi$ with general $\pi$.

**Definition 8.** Fix a tuple $\pi = (\pi_1, \ldots, \pi_n)$ of linear orders on $\{0,1\}$. Its *sign vector* is $\tau = (\tau_1, \ldots, \tau_n) \in \{\pm 1\}^n$, where $\tau_i = 1$ if $0 <_{\pi_i} 1$ and $\tau_i = -1$ if $1 <_{\pi_i} 0$.

**Lemma 9.** *Fix a tuple of linear orders $\pi$ with sign vector $\tau$. Then a function $f \colon \{0,1\}^n \to \mathbb{R}$ is $\pi$-supermodular if and only if $\tau_i \tau_j A^{(ij)} f \geq 0$, for all $i, j \in [n]$ with $i \neq j$.*

**Example 10** (Three-bit supermodular functions). For $X = \{0,1\}^3$, we have $\frac{1}{2} 2^3 = 4$ supermodular cones $\mathcal{L}_\pi$, given by sign vectors $\tau \in \{\pm 1\}^3$ up to global sign change. Each cone is described by $\binom{3}{2} \times 2^1 = 6$ elementary imset inequalities, collected into three matrices $A^{(ij)} \in \mathbb{R}^{2 \times 8}$. By Lemma 9, the sign of the inequality depends on the product $\tau_i \tau_j$:

| $\tau$ | $A^{(12)}f$ | $A^{(13)}f$ | $A^{(23)}f$ |
|---|---|---|---|
| $(1,1,1)$ | $+$ | $+$ | $+$ |
| $(-1,1,1)$ | $-$ | $-$ | $+$ |
| $(1,-1,1)$ | $-$ | $+$ | $-$ |
| $(1,1,-1)$ | $+$ | $-$ | $-$ |

In Appendix H we show that supermodular cones have tiny relative volume, at most $(0.85)^{2^n}$.

## 3 SUPERMODULAR RANK

We describe the facet defining inequalities of Minkowski sums of $\pi$-supermodular cones. Given two cones $\mathcal{P}$ and $\mathcal{Q}$, their Minkowski sum $\mathcal{P} + \mathcal{Q}$ is the set of points $p + q$, where $p \in \mathcal{P}$ and $q \in \mathcal{Q}$. For a partial order $\pi$ with sign vector $\tau$, we sometimes write $\mathcal{L}_\tau$ for $\mathcal{L}_\pi$.

**Example 11** (Sum of two three-bit supermodular cones). We saw in Example 10 that there are 4 distinct $\pi$-supermodular cones $\mathcal{L}_\pi$, each defined by $\binom{3}{2}2^{3-2} = 6$ elementary imset inequalities. The inequalities defining the Minkowski sum $\mathcal{L}_{(1,1,1)} + \mathcal{L}_{(-1,1,1)}$ are $A^{(23)}f \geq 0$. That is, the facet inequalities of the Minkowski sum are those inequalities that hold on both individual cones.

We develop general results on Minkowski sums of cones and apply them to the case of supermodular cones in Appendix D. We show that the observation in Example 11 holds in general: sums of $\pi$-supermodular cones are defined by the facet inequalities that are common to all supermodular summands. In particular, the Minkowski sum is as large a set as one could expect.

**Theorem 12** (Facet inequalities of sums of supermodular cones). *Fix a tuple of partial orders $\pi^{(1)}, \ldots, \pi^{(m)}$. The Minkowski sum of supermodular cones $\mathcal{L}_{\pi^{(1)}} + \cdots + \mathcal{L}_{\pi^{(m)}}$ is a convex polyhedral cone whose facet inequalities are the facet defining inequalities common to all $m$ cones $\mathcal{L}_{\pi^{(i)}}$.*

*Proof idea.* Each $\mathcal{L}_{\pi^{(i)}}$ is defined by picking sides of a fixed set of hyperplanes. All supermodular cones are full dimensional. If two cones lie on the same side of a hyperplane, then so does their Minkowski sum. Hence the sum is contained in the cone defined by the facet defining inequalities common to all $m$ cones $\mathcal{L}_{\pi^{(i)}}$. We show that the sum fills this cone. This generalizes the fact that a full dimensional cone $\mathcal{P} \subseteq \mathbb{R}^d$ satisfies $\mathcal{P} + (-\mathcal{P}) = \mathbb{R}^d$. $\square$

**Definition 13.** The *supermodular rank* of a function $f\colon \{0,1\}^n \to \mathbb{R}$ is the smallest $r$ such that $f = f_1 + \cdots + f_r$, where each $f_i$ is a $\pi$-supermodular function for some $\pi$.

**Theorem 14** (Maximum supermodular rank). *For $n \geq 3$, the maximum supermodular rank of a function $f\colon \{0,1\}^n \to \mathbb{R}$ is $\lceil \log_2 n \rceil + 1$. Moreover, submodular functions in the interior of $-\mathcal{L}_{(1,\ldots,1)}$ have supermodular rank $\lceil \log_2 n \rceil$.*

*Proof idea.* We find $\lceil \log_2 n \rceil + 1$ $\pi$-supermodular cones whose Minkowski sum fills the space. To remove as many inequalities as possible, we choose cones that share as few inequalities as possible, by Theorem 12. The sign vectors should differ at about $n/2$ coordinates, by Lemma 9. A recursive argument shows that $\lceil \log_2 n \rceil + 1$ cones suffices. $\square$

Submodular functions $f$ in the interior of $-\mathcal{L}_{(1\ldots1)}$ do not have full rank. They satisfy $A^{(ij)}f < 0$. We believe that full rank functions $f$ do not satisfy $A^{(ij)}f > 0$ or $A^{(ij)}f < 0$, for any $i \neq j$.

**Example 15.** Any function $f\colon \{0,1\}^3 \to \mathbb{R}$ can be written as a sum of at most three $\pi$-supermodular functions. Furthermore, there are functions that cannot be written as a sum of two $\pi$-supermodular functions. Indeed, Example 10 shows that any two $\pi$-supermodular cones share one block of inequalities, and three do not share any inequalities.

Definition 13 has implications for implicit descriptions of certain probabilistic graphical models: models with a single binary hidden variable as well as restricted Boltzmann machines (RBMs). Specifically, a probability distribution in the RBM model with $r$ hidden variables has supermodular rank at most $r$. We discuss these connections in Appendix I.

**Example 16** (Poset of sums of three-bit supermodular cones). We have four supermodular cones, with $\tau = (1, 1, 1), (-1, -1, 1), (-1, 1, -1), (1, -1, -1)$. In this case, all sums of pairs and all sums of triplets of cones behave similarly, in the sense that they have the same number of 0's in the vector $\xi$ indexing the Minkowski sum. This is shown in Figure 1.

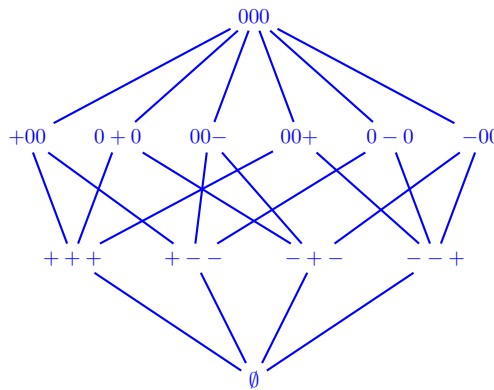

Figure 1: The inclusion poset of Minkowski sums of three-bit supermodular cones. The string in each node is the vector $\xi$ of signs of the inequalities for that supermodular cone or sum of supermodular cones.

## 3.1 ELEMENTARY SUBMODULAR RANK

We introduce a specialized notion of supermodular rank, which we call the elementary supermodular rank. This restricts to specific $\pi$. Later we focus on submodular functions, so we phrase the definition in terms of submodular functions.

**Definition 17.** If $\pi$ has a unique coordinate $i$ with the sign of $\pi_i$ equal to $-1$, then we call $-\mathcal{L}_\pi$ an *elementary submodular cone* and we say that a function $f \in -\mathcal{L}_\pi$ is $\{i\}$-submodular.

**Definition 18.** The *elementary submodular rank* of a function $f \colon \{0, 1\}^n \to \mathbb{R}$ is the smallest $r + 1$ such that $f = f_0 + f_{i_1} + \cdots + f_{i_r}$, where $f_0 \in -\mathcal{L}_{(1,\dots,1)}$ and $f_{i_j}$ is $\{i_j\}$-submodular.

Table 1: Volumes

| Submodular | | | |
|---|---|---|---|
| $n$ | rank 1 | rank 2 | rank 3 | rank 4 |
| 3 | 12.5% | 74.9% | 100% | - |
| 4 | 0.0072% | 5.9% | 100% | - |
| Elementary Submodular | | | |
| $n$ | rank 1 | rank 2 | rank 3 | rank 4 |
| 3 | 3.14% | 53.16% | 100% | - |
| 4 | $6 \cdot 10^{-4}$% | 0.38% | 29% | 100% |

**Theorem 19** (Maximum elementary submodular rank). *For $n \geq 3$, the maximum elementary submodular rank of a function $f \colon \{0, 1\}^n \to \mathbb{R}$ is $n$. Moreover, a supermodular function in the interior of $\mathcal{L}_{(1,\dots,1)}$ has elementary submodular rank $n$.*

The proof is given in Appendix D.4 using similar techniques to Theorem 14. The relative volume of functions of different ranks is shown in Table 1, with details in Appendix H.

**Low Elementary Submodular Rank Approximations** Given $f \colon \{0, 1\}^n \to \mathbb{R}$ and a target rank $r$, we seek an *elementary submodular rank-$r$ approximation* of $f$. This is a function $g \colon \{0, 1\}^n \to \mathbb{R}$ that minimizes $\|f - g\|_{\ell_2}$, with $g$ elementary submodular rank $r$. The set of elementary submodular rank $r$ functions is a union of convex cones, which is in general not convex. However, for fixed $\pi^{(1)}, \dots, \pi^{(r)}$, finding the closest point to $f$ in $\mathcal{L}_{\pi^{(1)}} + \cdots + \mathcal{L}_{\pi^{(r)}}$ is a convex problem. To find $g$, we compute the approximation for all rank $r$ convex cones and pick the function with the least error. Computing the projection onto each cone may be challenging, due to the number of facet-defining inequalities. We use PROJECT AND FORGET (Sonthalia and Gilbert, 2022), detailed in Appendix E.

## 4 SET FUNCTION OPTIMIZATION

The elementary submodular rank gives a gradation of functions. We apply it to set function optimization. Here we show that the theoretical guarantees that exist for submodular function optimization can applied to a broader family of functions. Namely, functions with low elementary submodular

rank. This extended applicability comes with increased running time. Specifically, the new runtime is exponential in $r$. However, we present a lower bound that shows that the added complexity is unavoidable. The first application is to constrained set function maximization. The second application to the ratio of set function minimization can be found in Appendix F.3.

**Definition 20.** A set function $f : 2^V \to \mathbb{R}$ is *monotone (increasing)* if $f(A) \leq f(B)$ for all $A \subseteq B$, *normalized* if $f(\emptyset) = 0$, and *positive* if $f(A) > 0$ for all $A \in 2^V \setminus \{\emptyset\}$.

Examples of monotone submodular functions include entropy $S \mapsto H(X_S)$ and mutual information $S \mapsto I(Y; X_S)$. Unconstrained maximization gives an optimum at $S = V$, but when constrained to subsets with upper bounded cardinality the problem is NP-hard. The cardinality constraint is a type of matroid constraint.

**Definition 21.** A system of sets $\mathcal{M} \subseteq 2^V$ is a *matroid*, if (i) $S \in \mathcal{M}$ and $T \subset S$ implies $T \in X$ and (ii) $S, T \in \mathcal{M}$ and $|T| + 1 = |S|$ implies $\exists e \in S \setminus T$ such that $T \cup \{e\} \in \mathcal{M}$. The *matroid rank* is the cardinality of the largest set in $\mathcal{M}$.

---

**Algorithm 1** GREEDY

1: **function** GREEDY($f, \mathcal{M}$)
2: $S_0 = \emptyset$, $F = \{e \in V : S_0 \cup \{e\} \in \mathcal{M}\}$
3: **while** $F \neq \emptyset$ **do**
4:  $e = \arg\max_{e \in F} \Delta(e|S_k)$
5:  $S_{k+1} = S_k \cup \{e\}$
6:  $F = \{e \in V : S_{k+1} \cup \{e\} \in \mathcal{M}\}$
7: **end while**
8: **return** $S_k$
9: **end function**

---

**Problem 22.** *Let $f$ be a monotone increasing normalized set function and let $\mathcal{M} \subseteq 2^V$ be a matroid. The $\mathcal{M}$-matroid constrained maximization problem is $\max_{S \in \mathcal{M}} f(S)$. The cardinality constraint problem is the special case $\mathcal{M} = \{S \subseteq V : |S| \leq m\}$, for a given $m \leq |V|$.*

A natural approach to finding an approximate solution to cardinality constrained monotone set function optimization is by an iteration that mimics gradient ascent. Given $S \subseteq V$ and $e \in V$, the discrete derivative of $f$ at $S$ in direction $e$ is

$$\Delta(e|S) = f(S \cup \{e\}) - f(S).$$

The GREEDY algorithm (Nemhauser et al. (1978)) produces a sequence of sets starting with $S_0 = \emptyset$, adding at each iteration an element $e$ that maximizes the discrete derivative subject to $S_i \cup \{e\} \in \mathcal{M}$ until no further additions are possible, see Algorithm 1.

**Submodular functions.** A well-known result by Nemhauser et al. (1978) shows that GREEDY has an approximation ratio (i.e., returned value divided by optimum value) of at least $(1 - e^{-1})$ for monotone submodular maximization with cardinality constraints. Călinescu et al. (2011); Filmus and Ward (2012) show a polynomial time algorithm exists for the matroid constraint case with the same approximation ratio of $(1 - e^{-1})$. These guarantees can be refined by measuring how far a submodular function is from being modular.

**Definition 23** (Conforti and Cornuéjols (1984)). The *total curvature* of a normalized, monotone increasing submodular set function $f$ is

$$\hat{\alpha} := \max_{e \in \Omega} \frac{\Delta(e|\emptyset) - \Delta(e|V \setminus \{e\})}{\Delta(e|\emptyset)}, \quad \text{where } \Omega = \{e \in V : \Delta(e|\emptyset) > 0\}.$$

It can be shown that $\hat{\alpha} = 0$ if and only if $f$ is modular, and that $\hat{\alpha} \leq 1$ for any submodular $f$. Building on this, Sviridenko et al. (2017) presents a NON-OBLIVIOUS LOCAL SEARCH GREEDY algorithm. They proved for any $\epsilon$ an approximation ratio of $(1 - \hat{\alpha}e^{-1} + O(\epsilon))$ for the matroid constraint and that this approximation ratio is optimal. See Appendix F.

**Approximately submodular functions.** To optimize functions that are not submodular, many prior works have looked at approximately submodular functions. We briefly discuss these here.

**Definition 24.** Fix a non-negative monotone set function $f : 2^V \to \mathbb{R}$.

- The *submodularity ratio* (Bian et al. (2017); Wang et al. (2019)) with respect to a set $X$ and integer $m$ is

$$\gamma_{X,m} := \min_{T \subset X, S \subset V, |S| \leq m, S \cap T = \emptyset} \frac{\sum_{e \in S} \Delta(e|T)}{\Delta(S|T)}.$$

 We drop the subscripts when $X = V$ and $m = |V|$.
- The *generalized curvature* (Bian et al. (2017)) is the smallest $\alpha$ s.t. for all $T, S \in 2^V$ and $e \in S \setminus T$,

$$\Delta(e|(S \setminus \{e\}) \cup T) \geq (1 - \alpha)\Delta(e|S \setminus \{e\}).$$

**Remark 25.** For monotone increasing $f$, we have $\gamma \in [0,1]$, with $\gamma = 1$ if and only if $f$ is submodular as well as $\alpha \in [0,1]$ and $\alpha = 0$ iff $f$ is supermodular. Thus, if $\alpha = 0$ and $\gamma = 1$, then $f$ is modular. We compare these notions to the elementary submodular rank in Appendix F.6.

Bian et al. (2017) obtained a guarantee of $\frac{1}{\alpha}(1 - e^{-\alpha\gamma})$ for the GREEDY algorithm on the cardinality constraint problem. For optimization of non-negative monotone increasing $f$ with general matroid constraints, Buchbinder et al. (2014); Chen et al. (2018); Gatmiry and Gomez-Rodriguez (2018) provide approximation guarantees that depend on $\gamma$ and $\alpha$.

**Functions with bounded elementary rank.** We formulate an algorithm with guarantees for the optimization of set functions with bounded elementary submodular rank. We first give definitions and properties of low elementary submodular rank functions.

**Definition 26.** Given $A \subseteq B \subseteq V$, define $\Pi(A, B) := \{C \subseteq V : C \cap B = A\}$. Given a set function $f : 2^V \to \mathbb{R}$, we let $f_{A,B}$ denote its restriction to $\Pi(A, B)$.

---

**Algorithm 2** R-SPLIT

1: **function** R-SPLIT($f, r, \mathcal{A}$ - subroutine)
2:   **for** $A \subseteq B \subseteq V$ with $|B| = r$ **do**
3:     run $\mathcal{A}$ on $f_{A,B}$
4:   **end for**
5:   run $\mathcal{A}$ on $f$
6:   **return** Best seen set
7: **end function**

---

Note that $\Pi(A, B) \cong 2^{V \setminus B}$. If $B = \{i\}$, then $f_{\{i\}, B}$ and $f_{\emptyset, B}$ are the pieces of $f$ on the sets that contain (resp. do not contain) $i$. If $|B| = m$, then we have $2^m$ pieces, the choices of $A \subseteq B$.

**Proposition 27.** *A set function $f$ has elementary submodular rank $r + 1$, with decomposition $f = f_0 + f_{i_1} + \cdots + f_{i_r}$, if and only if $f_{A,B}$ is submodular for $B = \{i_1, \ldots, i_r\}$ and any $A \subseteq B$.*

The pieces $f_{A,B}$ behave well in terms of their submodularity ratio and curvature.

**Proposition 28.** *If $f$ is a set function with submodularity ratio $\gamma$ and generalized curvature $\alpha$, then $f_{A,B}$ has submodularity ratio $\gamma_{A,B} \geq \gamma$ and generalized curvature $\alpha_{A,B} \leq \alpha$.*

We propose the algorithm R-SPLIT, see Algorithm 2. The idea is to run GREEDY, or another optimization method, on the pieces $f_{A,B}$ separately and then choose the best subset.

**Definition 29.** We define

$$\alpha_r := \min_{B \subseteq V, |B| \leq r} \left( \max_{A \subseteq B} \alpha_{A,B} \right) \quad \text{and} \quad \gamma_r := \max_{B \subseteq V, |B| \leq r} \left( \min_{A \subseteq B} \gamma_{A,B} \right).$$

We can now state our main result.

**Theorem 30** (Guarantees for R-SPLIT). *Let $\mathcal{A}$ be an algorithm for matroid constrained maximization of set functions, such that for any monotone, non-negative function $g \colon 2^W \to \mathbb{R}$, $|W| = m$, with generalized curvature $\alpha$ and submodularity ratio $\gamma$, algorithm $\mathcal{A}$ makes $O(q(m))$ queries to the value of $g$, where $q$ is a polynomial, and returns a solution with approximation ratio $R(\alpha, \gamma)$. If $f \colon 2^{[n]} \to \mathbb{R}$ is a monotone, non-negative function with generalized curvature $\alpha$, submodularity ratio $\gamma$, and elementary submodular rank $r + 1$, then R-SPLIT with subroutine $\mathcal{A}$ runs in time $O(2^r n^r q(n))$ and returns a solution with approximation ratio $\max\{R(\alpha, \gamma), R(\alpha_r, 1)\}$.*

**Remark 31** (Approximation ratio). The approximation ratio $R(\alpha, \gamma)$ usually improves as $\gamma$ increases, as seen for the function $\frac{1}{\alpha}(1 - e^{-\alpha\gamma})$ from Bian et al. (2017). The case $\gamma = 1$ corresponds to the function being submodular. We split an elementary rank-$(r + 1)$ function into the appropriate $2^r$ pieces, then we run the subroutine $\mathcal{A}$ on submodular functions Hence obtaining the best available guarantees.

**Remark 32** (Time complexity). For fixed $r$ we give a polynomial time approximation algorithm for elementary submodular rank-$(r + 1)$ functions. If we assume knowledge of the cones involved in a decomposition of $f$, then we need only optimize over one split, and we can drop the time complexity factor $n^r$. With the extra information about the decomposition, this is a fixed parameter tractable (FPT) time $O(1)$ approximation for monotone function maximization, parameterized by the elementary submodular rank. This suggests that determining the cones in the decomposition may be a difficult problem.

In Table 2 we instantiate several corollaries of Theorem 30 for specific choices of the subroutine and compare them with the prior work mentioned above (see Appendix F for details). Table 2 should be read as follows. We have an approximation ratio for every row in the prior work section of the table.

We can extend that result to the case with an elementary low-rank function. For example, the first row in the table shows that for submodular functions with matroid constraints, there exists an $\tilde{O}(n^8)$ algorithm that results in a $1 - e^{-1}$ approximation. Note that this result is for submodular functions. We can then extend this for elementary submodular rank-$r + 1$ functions. This is the last line of the table. Here, we see that we pay the time penalty of $O(2^r n^r)$ to do so. In the table we only extend the best three results. We also provide a lower bound.

**Theorem 33** (Lower Bound). *Any deterministic procedure that achieves an $O(1)$ approximation ratio for maximizing elementary submodular rank-$(r + 1)$ set functions requires at least $2^r$ function queries.*

Table 2: Approximation ratios and time complexity for maximizing a monotone, normalized function $f$ on $2^{[n]}$ of total curvature $\hat{\alpha}$, generalized curvature $\alpha$, submodularity ratio $\gamma$, and elementary submodular rank $r + 1$. Here $m$ is the bound in the cardinality constraint, and $\rho$ is the matroid rank in the Matroid constraint. Results with $^*$ are in expectation over randomization in the algorithm.

| | Sub-modular | Low Rank | Card. Constr. | Matroid Constr. | Approximation Ratio | Time | Ref. |
|---|---|---|---|---|---|---|---|
| **Prior Work** | ✓ | - | - | ✓ | $1 - e^{-1}$ $^*$ | $\tilde{O}(n^8)$ | Călinescu et al. (2011) |
| | ✓ | - | ✓ | - | $1 - \hat{\alpha}e^{-1} - O(\epsilon)$ | $O(\epsilon^{-1}poly(n))$ | Sviridenko et al. (2017) |
| | - | - | ✓ | - | $\frac{1}{\alpha}(1 - e^{-\alpha\gamma})$ | $O(nm)$ | Bian et al. (2017) |
| | - | - | - | ✓ | $(1 + \gamma^{-1})^{-2}$ | $O(n^2)$ | Chen et al. (2018) |
| | - | - | - | ✓ | $\frac{0.4\gamma^2}{\sqrt{\rho\gamma}+1}$ | $O(n^2)$ | Gatmiry and Gomez-Rodriguez (2018) |
| | - | - | - | ✓ | $(1+(1-\alpha)^{-1})^{-1}$ | $O(n^2)$ | Gatmiry and Gomez-Rodriguez (2018) |
| **This Work** | - | ✓ | ✓ | - | $1 - \hat{\alpha}e^{-1} - O(\epsilon)$ | $O(\epsilon^{-1}2^r n^r \cdot poly(n))$ | Cor 67 |
| | - | ✓ | ✓ | - | $\frac{1}{\alpha_r}(1 - e^{-\alpha_r})$ | $O(2^r n^r \cdot nm)$ | Cor 68 |
| | - | ✓ | - | ✓ | $1 - e^{-1}$ $^*$ | $\tilde{O}(2^r n^r \cdot n^8)$ | Cor 69 |

## 5 EXPERIMENTS

Our theory shows that we can extend theoretical guarantees that exist for submodular function optimization to low elementary submodular rank optimization. We take set functions that are not submodular, run our new algorithm, and show an improved performance. We consider four types of functions: DETERMINANTAL, BAYESIAN, COLUMN, and RANDOM, detailed in Appendix G.

**Submodularity ratio and generalized curvature.** We compute $\alpha_r$ and $\gamma_r$ and the resulting approximation ratio guarantee for constrained maximization for functions of the four different types, for $n = 8$ and log number of pieces $0 \leq r \leq 4$. We report the mean and standard error for 50 functions in each case. Figure 2a shows that the approximation ratio guarantee can increase quickly: for DETERMINANTAL it improves by 400% by $r = 4$.

**Low elementary rank approximations.** We compute low elementary submodular rank approximations for $n = 7$ and $1 \leq r + 1 \leq 7$. Figure 2b shows the relative error $\|f - g\|_{\ell_2}/\|f\|_{\ell_2}$ and Figure 2c the running times (see Appendix E). We see that COLUMN has a low elementary submodular rank, $r + 1 = 4$. The computation time peaks near $r + 1 = n/2$ and decreases for larger $r + 1$ as there are fewer Minkowski sums and fewer constraints.

**r-Split Greedy with small $n$.** We evaluate the improvement that R-SPLIT provides over GREEDY. Let $n = 20$ and maximize functions with a cardinality constraint $m = 10$. Figure 3a shows the approximation ratio for GREEDY and R-SPLIT GREEDY for $r = 1, 2, 3$, as well as how often the

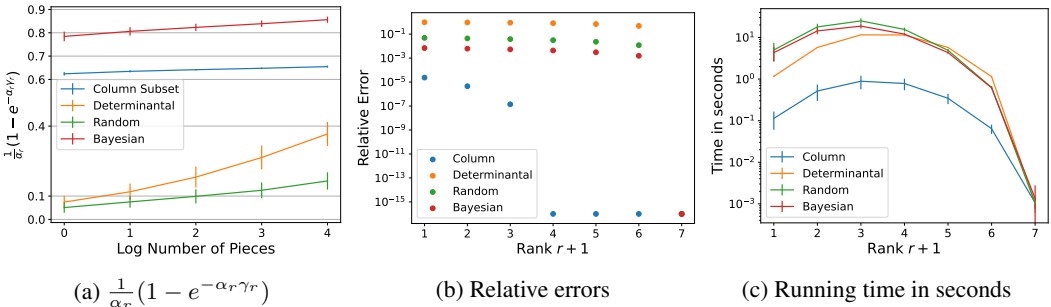

(a) $\frac{1}{\alpha_r}(1 - e^{-\alpha_r \gamma_r})$      (b) Relative errors      (c) Running time in seconds

Figure 2: Shown are (a) how the bound from Bian et al. (2017) changes when we use $\alpha_r$ and $\gamma_r$ (see Definition 29); higher values correspond to better guarantees, (b) the relative error when approximating a function by an elementary rank-$(r + 1)$ function, and (c) the running times for computing the approximation.

optimal solution was found. All four algorithms outperform their theoretical bounds. In all cases, increments in $r$ increase the percentage of times the (exact) optimal solution is found.

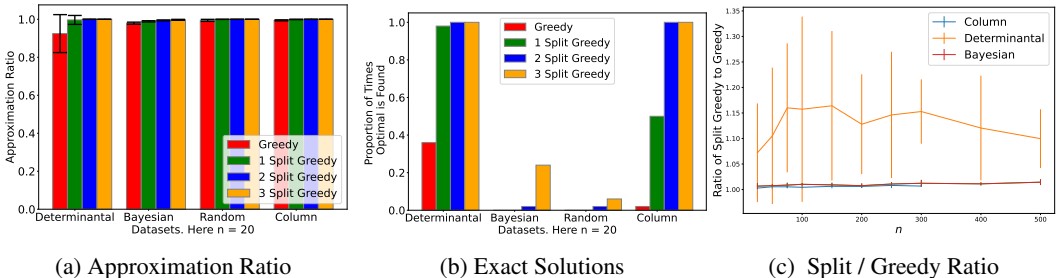

(a) Approximation Ratio      (b) Exact Solutions      (c) Split / Greedy Ratio

Figure 3: Shown are (a) the approximation ratio for the solution found versus the optimal solution averaged over 50 objective functions in each type, (b) the proportion of instances for which the algorithms find the optimal solution, (c) the ratio of the solution found by 1 SPLIT and GREEDY.

**r-Split Greedy with large $n$.** We now consider larger values of $n$, with a maximum cardinality of 15. Since we do not know the optimal solution, in Figure 3c we plot the ratio of R-SPLIT GREEDY to GREEDY. For BAYESIAN our approach improves the quality of the solution found by about 1% and for DETERMINANTAL by 5–15% on average. Running the experiment for RANDOM is not viable as we cannot store it for oracle access. Computing the COLUMN function takes longer than computing DETERMINANTAL and BAYESIAN. Hence we only ran it until $n = 300$. For $n = 300$, there are $\binom{300}{15} \approx 8 \times 10^{24}$ possible solutions.

## 6 CONCLUSION

We introduced the notion of submodular and supermodular rank for set functions along with geometric characterizations. Based on this we developed algorithms for constrained set function maximization and ratios of set functions minimization with theoretical guarantees improving several previous guarantees for these problems. Our algorithms do not require knowledge of the rank decomposition and show improved empirical performance on several commonly considered tasks even for small choices of $r$. For large $n$ it becomes unfeasible to evaluate all splits for large $r$, and one could consider evaluating only a random selection. It will be interesting to study in more detail the rank of typical functions. The theoretical complexity and practical approaches for computing rank decompositions remain open problems with interesting consequences. Another natural extension of our work is to consider general lattices, involving non-binary variables.

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

## A  NOTATION

We summarize our notation in the following table.

| | |
|---|---|
| $[n]$ | $\{1, \ldots, n\}$. |
| $(X, \pi)$ | A poset $X$ with partial order defined by a tuple $\pi$. |
| $f$ | A function from a poset $(X, \pi)$ to $\mathbb{R}$. |
| $\mathcal{L}_\pi$ | The cone of $\pi$-supermodular functions. |
| $\pi$ | Tuple of $n$ linear orders $\pi = (\pi_1, \ldots, \pi_n)$, see Definition 3. |
| $A^{(ij)}$ | The $(ij)$ elementary imset matrix, a $2^{n-2} \times 2^n$ matrix that collects the imset inequalities for each $x \in \{0,1\}^{n-2}$, see Definition 5. |
| $\tau$ | Sign vector $\tau = (\tau_1, \ldots, \tau_n)$ of a tuple of linear orders $\pi$, see Definition 8. |
| $\xi$ | Vector in $\{-1, 0, 1\}^{\binom{n}{2}}$, described in Definition 42. |

Notions of curvature and submodularity ratio.

| | |
|---|---|
| $\hat{\alpha}$ | The *total curvature* of a normalized, monotone increasing submodular function $f$ is $\hat{\alpha} := \max_{e \in \Omega} \frac{\Delta(e\|\emptyset) - \Delta(e\|V \setminus \{e\})}{\Delta(e\|\emptyset)}$, where $\Omega = \{e \in V : \Delta(e\|\emptyset) > 0\}$ |
| $\gamma_{X,m}$ | The *submodularity ratio* of a non-negative monotone function with respect to a set $X$ and integer $m$ is $\gamma_{X,m} := \min_{T \subset X, S \subset V, \|S\| \le m, S \cap T = \emptyset} \frac{\sum_{e \in S} \Delta(e\|T)}{\Delta(S\|T)}$. Subscripts dropped for $X = V$, $m = k$ |
| $\alpha$ | The *generalized curvature* of a non-negative monotone set function is the smallest $\alpha$ s.t. for all $T, S \in 2^V$ and $e \in S \setminus T$, $\Delta(e\|(S \setminus \{e\}) \cup T) \ge (1-\alpha)\Delta(e\|S \setminus \{e\})$ |
| $\tilde{\alpha}$ | The *generalized inverse curvature* of a non-negative set function $f$ is the smallest $\tilde{\alpha}^f$ s.t. for all $T, S \in 2^V$ and $e \in S \setminus T$, $\Delta(e\|S \setminus \{e\}) \ge (1-\tilde{\alpha}^f)\Delta(e\|(S \setminus \{e\}) \cup T)$. |
| $\hat{c}$ | The *curvature* of $f$ with respect to $X$ is $\hat{c}^f(X) := 1 - \frac{\sum_{e \in X}(f(X) - f(X \setminus \{e\}))}{\sum_{e \in X} f(\{e\})}$ |

## B  BACKGROUND ON POSETS

We introduce relevant background for posets and partial orders.

**Definition 34.** Let $(X, \preceq)$ be a partially ordered set (poset). Given two elements $x, y \in X$,

1. The *greatest lower bound* $x \wedge y$ is a $z \in X$ such that $z \prec x, y$ and $w \preceq z$ for all $w \prec x, y$.

2. The *least upper bound* $x \vee y$ is a $z \in X$ such that $x, y \prec z$ and $z \preceq w$ for all $x, y \prec w$.

Posets such that any two elements have a least upper bound and a greatest lower bound are called *lattices*.

**Example 35.**

1. Let $X$ be the power set of some set, with $\prec$ the inclusion order. Given $x, y \in X$, we have $x \wedge y = x \cap y$ and $x \vee y = x \cup y$.

2. Let $X = [d_1]' \times [d_2]' \times \cdots \times [d_n]'$, where $[n]' := \{0, 1, 2, 3, \ldots, n-1\}$. Fix $x = (x_1, \ldots x_n)$ and $y = (y_1, \ldots, y_n)$ in $X$. Let $x \prec y$ if and only if $x \ne y$ and $x_i \le y_i$, for all $i = 1, \ldots, n$, where $\le$ is the usual ordering on natural numbers. Then $x \wedge y = (\min(x_1, y_1), \ldots, \min(x_n, y_n))$ and $x \vee y = (\max(x_1, y_1), \ldots, \max(x_n, y_n))$.

3. Let $X = X_1 \times \cdots \times X_n$, where $(X_i, \le_i)$ are linearly ordered spaces for $i = 1, \ldots, n$. For $x, y \in X$, let $x \prec y$ if and only if $x_i \le_i y_i$, for $i = 1, \ldots, n$ and $x \ne y$, where $\le_i$ is the linear ordering on $X_i$. Then $x \wedge y = (\min_{\le_1}(x_1, y_1), \ldots, \min_{\le_n}(x_n, y_n))$ and $x \vee y = (\max_{\le_1}(x_1, y_1), \ldots, \max_{\le_n}(x_n, y_n))$.

## C    DETAILS ON SUPERMODULAR CONES

We give examples to illustrate the elementary imset inequality matrices from Definition 5.

**Example 36** (Elementary imset inequalities). Given $f\colon \{0,1\}^n \to \mathbb{R}$, the elementary imset inequalities are

$$f_{\ldots 0 \ldots 1 \ldots} + f_{\ldots 1 \ldots 0 \ldots} \leq f_{\ldots 0 \ldots 0 \ldots} + f_{\ldots 1 \ldots 1 \ldots}, \tag{4}$$

where $f_{\ldots 0 \ldots 1 \ldots} := f(\cdots 0 \cdots 1 \cdots)$ and an index $(\cdots a \cdots b \cdots)$ has varying entries at two positions $i$ and $j$. Fixing $i$ and $j$, one has $2^{n-2}$ inequalities in equation 4. For example, if $i = 1$ and $j = 2$ then one obtains two inequalities:

$$f_{010} + f_{100} \leq f_{000} + f_{110}$$
$$f_{011} + f_{101} \leq f_{001} + f_{111}.$$

**Example 37** (Three-bit elementary imset inequality matrix). For $n = 3$, $A^{(12)}$ is the $2 \times 8$ matrix

$$A^{(12)} = \begin{array}{c} \\ 0 \\ 1 \end{array} \begin{array}{c} 000 \quad 001 \quad 010 \quad 011 \quad 100 \quad 101 \quad 110 \quad 111 \\ \left( \begin{array}{cccccccc} 1 & & -1 & & -1 & & 1 & \\ & 1 & & -1 & & -1 & & 1 \end{array} \right). \end{array}$$

We prove Lemma 9, which describes the $\pi$-supermodular cones in terms of signed elementary imset ienqualities:

**Lemma 9.** *Fix a tuple of linear orders $\pi$ with sign vector $\tau$. Then a function $f\colon \{0,1\}^n \to \mathbb{R}$ is $\pi$-supermodular if and only if $\tau_i \tau_j A^{(ij)} f \geq 0$, for all $i, j \in [n]$ with $i \neq j$.*

*Proof.* The result is true by definition if $\tau = (1, \ldots, 1)$. Fix two binary vectors $x$ and $y$. If $\tau_i = 1$, the greatest lower bound $x \wedge y$ is a binary vector with $\min(x_i, y_i)$ at position $i$, while the least upper $x \vee y$ bound has $\max(x_i, y_i)$ at position $i$. If $\tau_i = -1$, then the greatest lower bound $x \wedge y$ has $\max(x_i, y_i)$ at position $i$, while $x \vee y$ has $\min(x_i, y_i)$ at position $i$. Fix $n = 2$ and assume $\tau = (-1, 1)$. Then $(00) \wedge (11) = (01)$ and $(00) \vee (11) = (10)$. Hence the supermodular inequality equation 1 applies to $x = (00)$ and $y = (11)$ to give

$$f_{00} + f_{11} \leq f_{01} + f_{10}.$$

For general $n$, assume that $(\tau_i, \tau_j) = (-1, 1)$ with (without loss of generality) that $i < j$. Then

$$f_{\ldots 0 \ldots 0 \ldots} + f_{\ldots 1 \ldots 1 \ldots} \leq f_{\ldots 0 \ldots 1 \ldots} + f_{\ldots 1 \ldots 0 \ldots}.$$

This is equation 4 with the sign of the inequality reversed. That is, with this partial order, the inequalities involving positions $i$ and $j$ are those of $-A^{(ij)}$. If $\tau = (-1, -1)$ then $(01) \wedge (10) = (00)$ and $(01) \vee (10) = (11)$ and there is no change in sign to the inequalities $A^{(ij)}$. $\square$

**Example 38** (Four-bit supermodular functions). For four binary variables, there are $2^{4-1} = 8$ distinct supermodular comes $\mathcal{L}_\pi$, given by a sign vector $\tau \in \{\pm 1\}^4$ up to global sign change, namely $(1,1,1,1)$, $(-1,1,1,1)$, $(1,-1,1,1)$, $(1,1,-1,1)$, $(1,1,1,-1)$, $(-1,-1,1,1)$, $(-1,1,-1,1)$, $(-1,1,1,-1)$. Each cone is described by $\binom{4}{2} \times 2^2 = 24$ elementary imset inequalities, collected into $\binom{4}{2}$ matrices $A^{(ij)} \in \mathbb{R}^{4 \times 16}$, where the sign of the inequality depends on the product $\tau_i \tau_j$. We give the signs of the inequalities for three sign vectors.

| $\tau$ | $A^{(12)}f$ | $A^{(13)}f$ | $A^{(14)}f$ | $A^{(23)}f$ | $A^{(24)}f$ | $A^{(34)}f$ |
|---|---|---|---|---|---|---|
| $(1,1,1,1)$ | + | + | + | + | + | + |
| $(-1,1,1,1)$ | − | − | − | + | + | + |
| $(-1,-1,1,1)$ | + | − | − | − | − | + |

## D    DETAILS ON SUPERMODULAR RANK

We provide details and proofs for the results in Section 3.

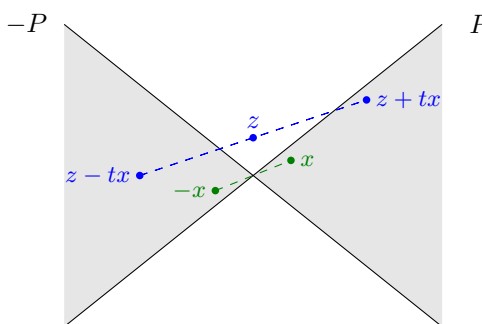

Figure 4: Proof of Proposition 39.

### D.1 FACET INEQUALITIES OF MINKOWSKI SUMS

We prove elementary properties of Minkowski sums of polyhedral cones. If two cones lie on the same side of a hyperplane through the origin, then so does their Minkowski sum, since $u^\top x_1 \geq 0$ and $u^\top x_2 \geq 0$ implies $u^\top(x_1 + x_2) \geq 0$. Moreover, the Minkowski sum of two convex cones $\mathcal{P}_1$ and $\mathcal{P}_2$ is convex, since

$$\mu(x_1 + x_2) + (1 - \mu)(y_1 + y_2) = (\mu x_1 + (1 - \mu)y_1) + (\mu x_2 + (1 - \mu)y_2)$$

holds, for $x_i, y_i \in \mathcal{P}_i$. Given a polyhedral cone $\mathcal{P}$ defined by inequalities $Ax \geq 0$, we denote by $-\mathcal{P}$ the cone defined by the inequalities $Ax \leq 0$. We write $Av > 0$ if every entry of the vector $Av$ is strictly positive.

**Proposition 39.** *Let $\mathcal{P} \subseteq \mathbb{R}^n$ be a full-dimensional polyhedral cone. Then $\mathcal{P} + (-\mathcal{P}) = \mathbb{R}^n$.*

*Proof.* Let $\mathcal{P} = \{v \in \mathbb{R}^n : Av \geq 0\}$. There exists $v \in \mathcal{P}$ with $Av > 0$, since $\mathcal{P}$ is full dimensional. Fix $z \in \mathbb{R}^n$. For sufficiently large $t$, we have $A(z + tv) \geq 0$ and $A(z - tv) \leq 0$. Hence $z + tv \in \mathcal{P}$ and $z - tv \in -\mathcal{P}$. We have $z = \frac{1}{2}(z + tv) + \frac{1}{2}(z - tv)$. Since $\mathcal{P}$ is closed under scaling by positive scalars, the first summand lies in $\mathcal{P}$ and the second in $-\mathcal{P}$. Hence $z$ is in the Minkowski sum. For a pictorial proof, see Figure 4. □

**Proposition 40.** *Given matrices $A_i \in \mathbb{R}^{n_i \times N}$, fix the two polyhedral cones*

$$\mathcal{P}_1 = \{v \in \mathbb{R}^N : A_1 v \geq 0, A_2 v \geq 0, A_3 v \geq 0\}$$

$$\mathcal{P}_2 = \{v \in \mathbb{R}^N : A_1 v \geq 0, A_2 v \leq 0, A_4 v \geq 0\}.$$

*Assume that the set*

$$S = \{v \in \mathbb{R}^N : A_1 v = 0, A_2 v > 0, A_3 v > 0, A_4 v < 0\}$$

*is non-empty. Then $\mathcal{P}_1 + \mathcal{P}_2 = \{v \in \mathbb{R}^N : A_1 v \geq 0\}$.*

*Proof.* Let $z \in \mathcal{P}_1 + \mathcal{P}_2$. Then $z = v + u$ for some $v \in \mathcal{P}_1$, and $u \in \mathcal{P}_2$. Hence $A_1 z = A_1(v + u) \geq 0$, so $\mathcal{P}_1 + \mathcal{P}_2 \subseteq \{z : A_1 z \geq 0\}$. For the reverse containment, take $z$ such that $A_1 z \geq 0$. Fix $v^* \in S$. There exists some $\lambda_1 \in \mathbb{R}_{\geq 0}$ such that for all $\lambda > \lambda_1$, we have $A_2(z + \lambda v^*) \geq 0$ and $A_3(z + \lambda v^*) \geq 0$. Then $z + \lambda v^* \in \mathcal{P}_1$, since

$$A_1(z + \lambda v^*) = A_1 z + \lambda A_1 v^* = A_1 z \geq 0.$$

Moreover, there exists some $\lambda_2 \in \mathbb{R}_{\geq 0}$ such that for all $\lambda > \lambda_2$, we have that $z - \lambda v^*$ satisfies

$$A_1(z - \lambda v^*) \geq 0, A_2(z - \lambda v^*) \leq 0 \text{ and } A_4(z - \lambda v^*) \geq 0,$$

so $z - \lambda v \in P_2$. Taking $\lambda > \max(\lambda_1, \lambda_2)$, we can write $z = \frac{1}{2}(z + \lambda v^*) + \frac{1}{2}(z - \lambda v^*)$ to express $z$ as an element of $\mathcal{P}_1 + \mathcal{P}_2$. □

### D.2 FACET INEQUALITIES FOR SUMS OF SUPERMODULAR CONES

**Example 41** (Sums of two four-bit supermodular cones)**.** We saw in Example 38 that there are eight possible $\pi$-supermodular cones $\mathcal{L}_\pi$, each defined by $\binom{4}{2}2^{4-2} = 24$ elementary imset inequalities. Here we study the facet inequalities for their Minkowski sums. The inequalities defining the Minkowski sum $\mathcal{L}_{(1,1,1,1)} + \mathcal{L}_{(-1,1,1,1)}$ are

$$A^{(23)}f \geq 0, \quad A^{(24)}f \geq 0, \quad A^{(34)}f \geq 0,$$

as can be computed using polymake Assarf et al. (2017). That is, the facet defining inequalities of the Minkowski sum are those that hold on both individual cones, and no others. We similarly compute the inequalities that define the Minkowski sum $\mathcal{L}_{(1,1,1,1)} + \mathcal{L}_{(-1,-1,1,1)}$. We obtain

$$A^{(12)}f \geq 0, \quad A^{(34)}f \geq 0.$$

Again, the Minkowski sum is described by just the inequalities present in both cones individually. Notice that $\mathcal{L}_{(1,1,1,1)} + \mathcal{L}_{(-1,1,1,1)}$ is defined by 12 inequalities while $\mathcal{L}_{(1,1,1,1)} + \mathcal{L}_{(-1,-1,1,1)}$ is defined by eight.

We show that the assumptions of Proposition 40 hold for sums of supermodular cones.

**Definition 42.** Given $\xi \in \{-1, 0, 1\}^{\binom{n}{2}}$, define $\mathcal{L}_\xi$ to be

$$\mathcal{L}_\xi = \{x \colon \xi_{ij}A^{(ij)}x \geq 0, \quad \text{for all } i \neq j\},$$

where $A^{(ij)}$ is the $(i, j)$ elementary imset matrix from Definiton 5. Similarly, define

$$\mathcal{L}'_\xi = \mathcal{L}_\xi \cap \{x \colon A^{(ij)}x = 0 \text{ for all } i \neq j \text{ with } \xi_{ij} = 0\},$$

The cones $\mathcal{L}_\xi$ and $\mathcal{L}'_\xi$ are $\pi$-supermodular cones in the special cases that $\xi_{ij} = \tau_i\tau_j$ for some $\tau \in \{\pm 1\}^n$.

**Lemma 43.** Fix $\xi \in \{-1, 0, 1\}^{\binom{n}{2}}$. There exists $z \in \mathcal{L}'_\xi$ with $\xi_{ij}A^{(ij)}z > 0$ for all $\xi_{ij} \neq 0$.

*Proof.* First assume just one entry $\xi_{ij}$ of $\xi$ is non-zero. Let $z^{(ij)} \in \mathbb{R}^{\{0,1\}^n}$ have entries $z_\ell^{(ij)} = c_{\ell_i\ell_j}$, for $\ell = (\ell_1, \ldots, \ell_n) \in \{0, 1\}^n$. Then all rows of $\xi_{ij}A^{(ij)}z^{(ij)}$ equal $\xi_{ij}(c_{00} + c_{11} - c_{01} - c_{10})$. We choose the four entries $c_{00}, c_{01}, c_{10}, c_{11}$ so that $\xi_{ij}(c_{00} + c_{11} - c_{01} - c_{10}) > 0$. Moreover, $A^{(i'j')}z^{(ij)}$ is zero for all other $\{i', j'\}$, since the value of $z_\ell^{(ij)}$ only depends on $\ell_i, \ell_j$. We conclude by setting $z = \sum z^{(ij)}$, where the sum is over $(i, j)$ with $\xi_{ij} \neq 0$. $\square$

**Proposition 44.** Fix $\xi^{(1)}, \xi^{(2)} \in \{-1, 0, 1\}^{\binom{n}{2}}$. The Minkowski sum $\mathcal{L}_{\xi^{(1)}} + \mathcal{L}_{\xi^{(2)}}$ is cut out by the inequalities common to both summands. That is, $\mathcal{L}_{\xi^{(1)}} + \mathcal{L}_{\xi^{(2)}} = \mathcal{L}_\xi$, where

$$\xi_{ij} = \begin{cases} \xi_{ij}^{(1)} & \xi_{ij}^{(1)} = \xi_{ij}^{(2)} \\ 0 & \text{otherwise.} \end{cases}$$

*Proof.* Define $\xi'$ by

$$\xi'_{ij} = \begin{cases} 0, & \xi_{ij}^{(1)} \cdot \xi_{ij}^{(2)} = 1 \\ \xi_{ij}^{(1)} & \xi_{ij}^{(1)} \cdot \xi_{ij}^{(2)} = -1 \\ \xi_{ij}^{(1)} & \xi_{ij}^{(1)} \neq 0, \xi_{ij}^{(2)} = 0 \\ -\xi_{ij}^{(2)} & \xi_{ij}^{(2)} \neq 0, \xi_{ij}^{(1)} = 0. \end{cases}$$

There exists $v$ in the interior of $\mathcal{L}'_{\xi'}$, by Lemma 43. This $v$ satisfies the assumption from Proposition 40 for the polyhedral cones $\mathcal{P}_1 = \mathcal{L}_{\xi^{(1)}}$ and $\mathcal{P}_2 = \mathcal{L}_{\xi^{(2)}}$. The four cases in the definition of $\xi'$ are the four cases $A_1v = 0, A_2v > 0, A_3v > 0, A_4v < 0$ in set $S$ of Proposition 40. $\square$

We can now show Theorem 12:

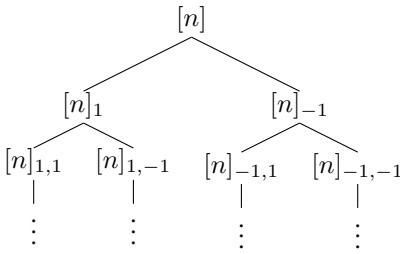

Figure 5: Illustration of the proof of Theorem 14.

**Theorem 12** (Facet inequalities of sums of supermodular cones). *Fix a tuple of partial orders* $\pi^{(1)}, \dots, \pi^{(m)}$. *The Minkowski sum of supermodular cones* $\mathcal{L}_{\pi^{(1)}} + \cdots + \mathcal{L}_{\pi^{(m)}}$ *is a convex polyhedral cone whose facet inequalities are the facet defining inequalities common to all* $m$ *cones* $\mathcal{L}_{\pi^{(i)}}$.

*Proof.* Let $\tau^{(s)}$ be the sign vector of $\pi^{(s)}$. By Proposition 44, the Minkowski sum in the statement is $\mathcal{L}_\xi$, where

$$\xi_{ij} = \begin{cases} \xi_{ij}^{(1)} & \tau_i^{(1)}\tau_j^{(1)} = \cdots = \tau_i^{(m)}\tau_j^{(m)} \\ 0 & \text{otherwise.} \end{cases}$$

$\square$

### D.3 MAXIMUM SUPERMODULAR RANK

We now prove Theorem 14.

**Theorem 14** (Maximum supermodular rank). *For* $n \geq 3$, *the maximum supermodular rank of a function* $f\colon \{0,1\}^n \to \mathbb{R}$ *is* $\lceil \log_2 n \rceil + 1$. *Moreover, submodular functions in the interior of* $-\mathcal{L}_{(1,\dots,1)}$ *have supermodular rank* $\lceil \log_2 n \rceil$.

*Proof.* The maximum supermodular rank is the minimal $m$ such that a union of cones of the form $\mathcal{L}_{\pi^{(1)}} + \cdots + \mathcal{L}_{\pi^{(m)}}$ fills the space of functions $f : \{0,1\}^n \to \mathbb{R}$. Let $\tau^{(k)}$ be the sign vector of partial order $\pi^{(k)}$. We show that the maximum supermodular rank is the smallest $m$ such that there exist partial orders $\pi^{(1)}, \dots \pi^{(m)}$ with no pair $i \neq j$ having the same value of the product of signs $\tau_i^{(k)} \cdot \tau_j^{(k)}$ for all $k = 1, \dots, m$. If there is no such pair $i, j$, then the Minkowski sum fills the space, by Lemma 9 and Theorem 12. Conversely, assume that $\xi_{ij} := \tau_i^{(k)} \cdot \tau_j^{(k)}$ is the same for all $k$, for some $i \neq j$. Let the other entries of $\xi$ be zero. Then the Minkowski sum $\mathcal{L}_{\pi^{(1)}} + \cdots + \mathcal{L}_{\pi^{(m)}}$ is contained in $\mathcal{L}_\xi$, by Theorem 12. A union of $\mathcal{L}_\xi$ with $\xi \neq 0$ cannot equal the whole space since $\xi_{ij} A^{(ij)} f \geq 0$ from Lemma 9 imposes $2^{n-2}$ inequalities and for $n \geq 3$ there exist functions with different signs for these two or more inequalities, since each inequality involves distinct indices. It, therefore, remains to study the sign vector problem.

Without loss of generality, let $\tau^{(1)} = (1, \dots, 1)$. Consider the partition $[n] = [n]_1 \cup [n]_{-1}$, where

$$\tau_i^{(2)} = \begin{cases} 1 & i \in [n]_1 \\ -1 & i \in [n]_{-1}. \end{cases}$$

Let $n_1 := |[n]_1|$. Of the $\binom{n}{2}$ pairs $i \neq j$ there are $n_1(n - n_1)$ with $\tau_i^{(2)}\tau_j^{(2)} = -1$. The quantity $n_1(n - n_1)$ is maximized when $n_1 = \frac{n}{2}$ (for $n$ even) or $n_1 = \frac{1}{2}(n \pm 1)$ (for $n$ odd). It remains to consider the pairs $i \neq j$ with $\tau_i^{(1)}\tau_j^{(1)} = \tau_i^{(2)}\tau_j^{(2)}$; that is, $(\tau_i^{(2)}, \tau_j^{(2)}) = (1,1)$ or $(-1,-1)$. We have reduced the problem to two smaller problems, each with $\binom{m}{2}$ pairs $i, j$ where $m \leq \frac{n+1}{2}$. We choose a partition of $[n]_1$ into two pieces, say $[n]_{1,1}$ and $[n]_{1,-1}$, and likewise for $[n]_{-1}$. Define $\tau^{(3)}$ to be 1 on $[n]_{1,1}, [n]_{-1,1}$ and $-1$ on $[n]_{1,-1}, [n]_{-1,-1}$. Then the pairs $i \neq j$ with $\tau_i^{(1)}\tau_j^{(1)} = \tau_i^{(2)}\tau_j^{(2)} = \tau_i^{(3)}\tau_j^{(3)}$ are those with $\{i, j\} \subset [n]_{ab}$ for some $a, b \in \{-1, 1\}$. In a sum of $m$ cones, the set $[n]$ has been divided into $2^{m-1}$ pieces. Hence there is one piece of size at least $\lceil \frac{n}{2^{m-1}} \rceil$, by the Pigeonhole

principle. This is at least two for $m \leq \lceil \log_2 n \rceil$. Conversely, choosing a splitting into two pieces of size as close as possible shows that for $m \geq \lceil \log_2 n \rceil + 1$ the set $[n]$ can be divided into pieces of size 1. $\qquad \square$

**Proposition 45** (Supermodular rank of submodular functions). *A strictly submodular function $f \colon 2^{[n]} \to \mathbb{R}$ has supermodular rank $\lceil \log_2 n \rceil$.*

*Proof.* We first show that there exists a submodular function $f$ of supermodular rank at least $\lceil \log_2 n \rceil$. Suppose that for all $f \in -\mathcal{L}_{(1,\dots,1)}$, the supermodular rank of $f$ was at most $\lceil \log_2 n \rceil - 1$. The sum $\mathcal{L}_{(1,\dots,1)} + (-\mathcal{L}_{(1,\dots,1)})$ is the whole space, by Proposition 39. Then the maximal supermodular rank would be $\lceil \log_2 n \rceil$, contradicting Theorem 14. Hence there exists $f \in -\mathcal{L}_{(1,\dots,1)}$ with supermodular rank at least $\lceil \log_2 n \rceil$.

A function $g$ in the interior of the submodular cone satisfies $A^{(ij)}g < 0$ for all $\{i,j\}$. Suppose that there exists such a $g$ with supermodular rank less than $\lceil \log_2 n \rceil$. Then there exist $\pi^{(1)}, \dots, \pi^{(m)}$ for $m < \lceil \log_2 n \rceil$, such that $g \in \mathcal{L}_{\pi^{(1)}} + \cdots + \mathcal{L}_{\pi^{(m)}}$. There exists $\xi \in \{-1, 0, 1\}^n$ such that

$$\mathcal{L}_\xi = \mathcal{L}_{\pi^{(1)}} + \cdots + \mathcal{L}_{\pi^{(m)}},$$

by Proposition 40. Hence $g$ satisfies inequalities $\xi_{ij} A^{(ij)} g \geq 0$, by Definition 42. Therefore $\xi_{ij} \in \{-1, 0\}$ for all $\{i,j\}$. It follows that all submodular functions are in $\mathcal{L}_\xi$, a contradiction since by the first paragraph of the proof there exist submodular $f$ of supermodular rank at least $\lceil \log_2 n \rceil$.

It remains to show that the supermodular rank of a submodular function is at most $\lceil \log_2 n \rceil$. That is, we aim to show that $\lceil \log_2 n \rceil$ cones can be summed to give some $\mathcal{L}_\xi$ with all $\xi_{ij}$ in $\{-1, 0\}$. In the proof of Theorem 14, we counted the number of supermodular cones that needed to give $\mathcal{L}_\xi$ with all $\xi_{ij} = 0$. Here, starting with a partial order with some $\tau_i \tau_j = -1$, instead of all $\tau_i \tau_j = 1$, shows that we require (at least) one fewer cone than in Theorem 14. $\qquad \square$

### D.4  Maximum Elementary Submodular Rank

A function can be decomposed as a sum of elementary submodular functions.

**Theorem 46.** *Let $f \colon \{0,1\}^n \to \mathbb{R}$. Then there exist $f_0, f_1, \dots, f_{n-1}$ with $f_0 \in -\mathcal{L}_{(1,\dots,1)}$ and $f_i \in -\mathcal{L}_{\tau^{(i)}}$ where $\tau_j^{(i)} = -1$ if and only if $j = i$, such that $f = f_0 + \sum_{i=1}^{n-1} f_i$.*

*Proof.* The sign vector $-\mathcal{L}_{(1,\dots,1)}$ is $\tau = (1, \dots, 1)$. For all $i,j$ we have $\tau_i \tau_j = 1$. For the cones $\mathcal{L}_{\tau^{(i)}}$, we have $\tau_i^{(i)} \tau_j^{(i)} = -1$ for all $j \neq i$. Hence there are no facet inequalities common to all $n$ cones. The result then follows from Proposition 44. $\qquad \square$

We can now prove Theorem 19:

**Theorem 19** (Maximum elementary submodular rank). *For $n \geq 3$, the maximum elementary submodular rank of a function $f \colon \{0,1\}^n \to \mathbb{R}$ is $n$. Moreover, a supermodular function in the interior of $\mathcal{L}_{(1,\dots,1)}$ has elementary submodular rank $n$.*

*Proof.* The maximum elementary submodular rank is at most $n$, by Theorem 46. For any $i_1, \dots, i_r$, the cone $-\mathcal{L}_{(1,\dots,1)} + (-\mathcal{L}_{\tau^{(i_1)}}) + \cdots + (-\mathcal{L}_{\tau^{(i_r)}})$, where $-\mathcal{L}_{\tau^{(i_j)}}$ is an $(i_j)$-th elementary submodular cone, is defined by the inequalities

$$A^{(jk)} f \leq 0, \quad j, k \notin \{i_1, \dots, i_r\},$$

see Proposition 40. This consists of $2^{n-2} \binom{n-r}{2}$ inequalities. A union of such functions therefore cannot equal the full space of functions if $r + 1 \leq n - 1$, as in the proof of Theorem 14.

If $f$ is in the interior of the super modular cone, then

$$A^{(ij)} h > 0, \quad \text{for all } i \neq j.$$

For $f$ to be in $-\mathcal{L}_{(1,\dots,1)} + (-\mathcal{L}_{\tau^{(i_1)}}) + \cdots + (-\mathcal{L}_{\tau^{(i_r)}})$, we need there to be no $\{i,j\}$ such that $\{i,j\} \cap \{i_1, \dots, i_r\} = \emptyset$. Thus, we require $r \geq n - 1$. $\qquad \square$

### D.5 INCLUSION RELATIONS OF SUMS OF SUPERMODULAR CONES

We briefly discuss the structure of the sets of bounded supermodular rank. Some tuples of super-modular cones are closer together than others in the sense that their Minkowski sum is defined by a larger number of inequalities, as we saw in Example 41.

We consider the sums $\mathcal{L}_{\pi^{(1)}} + \mathcal{L}_{\pi^{(2)}} + \cdots + \mathcal{L}_{\pi^{(m)}}$, for different choices of $(\pi^{(1)}, \ldots, \pi^{(m)})$ and $m$. This set can be organized in levels corresponding to the number of summands (rank) and partially ordered by inclusion. Each cone corresponds to a $\binom{n}{2}$-vector $\xi$ with entries indexed by $\{i,j\}$, $i \neq j$. A sign vector $\tau$ having $s$ entries $-1$ has $\xi$ vector with $s \cdot (n-s)$ entries $-1$.

**Example 47** (Poset of sums of three-bit supermodular cones). We have four supermodular cones, with $\tau = (1,1,1), (-1,-1,1), (-1,1,-1), (1,-1,-1)$. In this case, all sums of pairs and all sums of triplets of cones behave similarly, in the sense that they have the same number of $0$'s in the vector $\xi$ indexing the Minkowski sum. This is shown in Figure 6.

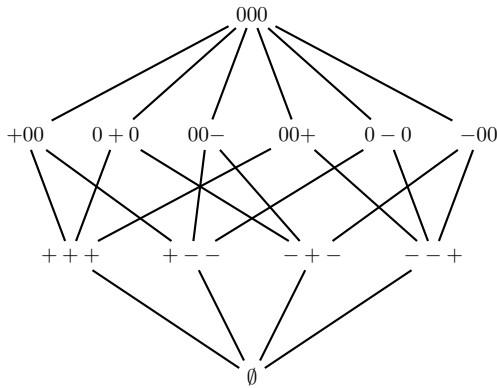

Figure 6: The inclusion poset of Minkowski sums of three-bit supermodular cones. The string in each node is the vector $\xi$ of signs of the inequalities for that supermodular cone or sum of super-modular cones. See Example 47.

**Example 48** (Poset of sums of four-bit supermodular cones). There are eight supermodular cones, given by the $\tau = (1,1,1,1), (-1,1,1,1), (1,-1,1,1), (1,1,-1,1), (1,1,1,-1), (-1,-1,1,1), (-1,1,-1,1), (-1,1,1,-1)$, each defining signs for the six columns $\{1,2\}, \{1,3\}, \{1,4\}, \{2,3\}, \{2,4\}, \{3,4\}$. In this case, we see that rank 2 nodes are not all equivalent, in the sense that they may have a different number of $0$'s (arbitrary sign). Some have three and some have four $0$'s. There are triplet sums that have all $0$'s, but not all do. See Figure 7.

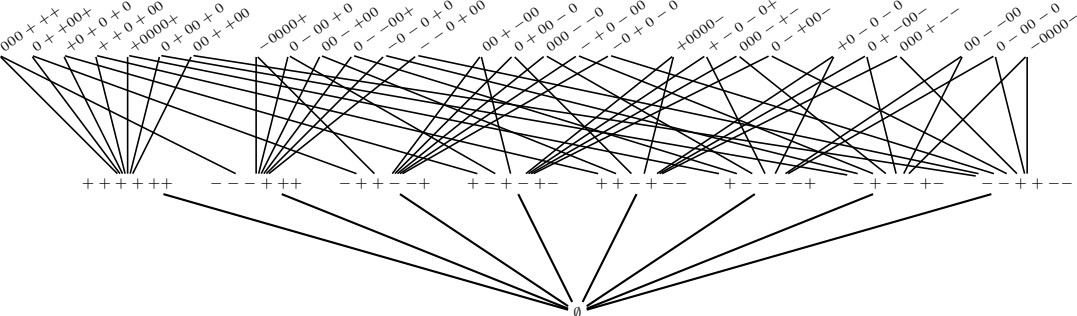

Figure 7: Inclusion poset of Minkowski sums of four-bit supermodular cones. There are 8 super-modular cones and 28 distinct Minkowski sums of pairs of supermodular cones. See Example 48.

### D.6 EXAMPLES OF LOW RANK FUNCTIONS

**RBM.**  The first example comes from Restricted Boltzmann Machines (RBMs). These are graphical models for modeling probability distributions on $\{0,1\}^n$. They have a hyperparameter $m$, the number of hidden nodes. Prior work (Allman et al., 2015) has shown that when $m = 1$, this model can represent distributions $f(x)$ if and only if $\log f$ is $\pi$-supermodular and satisfies certain polynomial equality constraints. Then, when we move to more significant values of $m$, it can be shown that the model can model distribution $f(x)$ only if $\log f$ is rank-$m$ supermodular. Thus $\log$ RBM distributions are functions with bounded supermodular rank. In this case, the optimization problem would correspond to finding modes of the distribution.

**One hidden layer neural networks**  Building on this, we have the following.

**Proposition 49.** *Let $f$ be a real-valued function on $\{0,1\}^n$ that is the composition of an affine function and a convex function. That is,*

$$f(x) = \phi(wx + c)$$

*where $w$ is a vector of length $1n$ and $c$ is a scalar and $\phi : \mathbb{R} \to \mathbb{R}$ is convex. Then $f$ is sign(w)-supermodular.*

*Proof.* We show that the elementary imset inequalities are satisfied. Such an inequality involves four vectors on a two-dimensional face of the cube $\{0,1\}^n$. We have two indices $x_i, x_j$ that vary and a fixed value of $x_{[n]\setminus\{i,j\}}$, the vector $x$ restricted to the set $[n] \setminus \{i, j\}$. Let $y$ be the entry of the face where $x_i = x_j = 0$. Letting $c' = c + Ay$, we seek to compare $\phi(c' + w_i) + \phi(c' + w_j)$ with $\phi(c') + \phi(c' + w_i + w_j)$. By the definition of sign(w)-supermodularity, if both entries $w_i$ and $w_j$ have the same sign, we require

$$\phi(c' + w_i) + \phi(c' + w_j) \le \phi(c') + \phi(c' + w_i + w_j)$$

while if $w_i w_j \le 0$, we require

$$\phi(c' + w_i) + \phi(c' + w_j) \ge \phi(c') + \phi(c' + w_i + w_j).$$

The inequalities hold by the fact that $\phi$ is convex. For example, if $w_i, w_j \ge 0$, then $c \le c + w_i, c + w_j \le c + w_i + w_j$, and we apply the definition of convexity as applied to a comparison of four points. □

Thus, in particular, a one-hidden layer ReLU neural network, when restricted to $\{0,1\}^n$ with $k$ hidden nodes with positive outer weights, is rank-$k$ supermodular

## E  DETAILS ON COMPUTING LOW RANK APPROXIMATIONS

The problem we are interested in is as follows. Given a target set function $f$ and partial orders $\pi^{(1)}, \ldots, \pi^{(m)}$ we minimize $\|f - g\|_2$ over $g \in \mathcal{L}_{\pi^{(1)}} + \cdots + \mathcal{L}_{\pi^{(m)}}$. While there are many algorithmic techniques that could solve this problem, we use PROJECT AND FORGET due to Sonthalia and Gilbert (2022), which is designed to solve highly constrained convex optimization problems and has been shown to be capable of solving problems with up to $10^{15}$ constraints. We first present a discussion of the method, adapted from Sonthalia and Gilbert (2022). Then we apply it to our problem.

### E.1  PROJECT AND FORGET

PROJECT AND FORGET is a conversion of Bregman's cyclic method into an active set method to solve metric constrained problems (Brickell et al. (2008); Fan et al. (2020); Gilbert and Sonthalia (2018)). It is an iterative method with three major steps per iteration: (i) (ORACLE) an (efficient) oracle to find violated constraints, (ii) (PROJECT) Bregman projection onto the hyperplanes defined by each of the active constraints, and (iii) (FORGET) the forgetting of constraints that no longer require attention. The main iteration with the above three steps is given in Algorithms 3. The PROJECT and FORGET functions are presented in Algorithm 4. To describe the details and guarantees we need a few definitions.

**Definition 50.** Given a convex function $f(x) : S \to \mathbb{R}$ whose gradient is defined on $S$, we define its *generalized Bregman distance* $D_f : S \times S \to \mathbb{R}$ as $D_f(x, y) = f(x) - f(y) - \langle \nabla f(y), x - y \rangle$.

**Definition 51.** A function $f : \Lambda \to \mathbb{R}$ is called a Bregman function if there exists a non-empty convex set $S$ such that $\overline{S} \subset \Lambda$ and the following hold:

(i) $f(x)$ is continuous, strictly convex on $\overline{S}$, and has continuous partial derivatives in $S$.

(ii) For every $\alpha \in \mathbb{R}$, the partial level sets $L_1^f(y, \alpha) := \{x \in \overline{S} : D_f(x, y) \leq \alpha\}$ and $L_2^f(x, \alpha) := \{y \in S : D_f(x, y) \leq \alpha\}$ are bounded for all $x \in \overline{S}, y \in S$.

(iii) If $y_n \in S$ and $\lim_{n \to \infty} y_n = y^*$, then $\lim_{n \to \infty} D_f(y^*, y_n) = 0$.

(iv) If $y_n \in S$, $x_n \in \overline{S}$, $\lim_{n \to \infty} D_f(x_n, y_n) = 0$, $y_n \to y^*$, and $x_n$ is bounded, then $x_n \to y^*$.

We denote the family of Bregman functions by $\mathcal{B}(S)$. We refer to $S$ as the zone of the function and we take the closure of the $S$ to be the domain of $f$. Here $\overline{S}$ is the closure of $S$.

This class of function includes many natural objective functions, including entropy $f(x) = -\sum_{i=1}^n x_i \log(x_i)$ with zone $S = \mathbb{R}_+^n$ (here $f$ is defined on the boundary of $S$ by taking the limit) and $f(x) = \frac{1}{p}\|x\|_p^p$ for $p \in (1, \infty)$. The $\ell_p$ norms for $p = 1, \infty$ are not Bregman functions but can be made Bregman functions by adding a quadratic term. That is, $f(x) = c^T x$ is a not Bregman function, but $c^T x + x^T Q x$ for any positive definite $Q$ is a Bregman function.

**Definition 52.** We say that a hyperplane $H_i$ is *strongly zone consistent* with respect to a Bregman function $f$ and its zone $S$, if for all $y \in S$ and for all hyperplanes $H$, parallel to $H_i$ that lie in between $y$ and $H_i$, the Bregman projection of $y$ onto $H$ lies in $S$ instead of in $\overline{S}$.

**Theorem 53** (Sonthalia and Gilbert (2022)). *If $f \in \mathcal{B}(S)$, $H_i$ are strongly zone consistent with respect to $f$, and $\exists x^0 \in S$ such that $\nabla f(x^0) = 0$, then*

1. *If the oracle returns each violated constraint with a positive probability, then any sequence $x^n$ produced by the above algorithm converges (with probability 1) to the optimal solution.*

2. *If $x^*$ is the optimal solution, $f$ is twice differentiable at $x^*$, and the Hessian $H := Hf(x^*)$ is positive definite, then there exists $\rho \in (0, 1)$ such that*

$$\lim_{\nu \to \infty} \frac{\|x^* - x^{\nu+1}\|_H}{\|x^* - x^\nu\|_H} \leq \rho \tag{5}$$

*where $\|y\|_H^2 = y^T H y$. The limit in equation 5 holds with probability 1.*

---

**Algorithm 3** General Algorithm.

---

1: **function** PROJECT AND FORGET($f$, $\mathcal{Q}$ - the oracle)
2:     $L^{(0)} = \emptyset$, $z^{(0)} = 0$. Initialize $x^{(0)}$ so that $\nabla f(x^{(0)}) = 0$.
3:     **while** Not Converged **do**
4:         $L = \mathcal{Q}(x^\nu)$
5:         $\tilde{L}^{(\nu+1)} = L^{(\nu)} \cup L$
6:         $x^{(\nu+1)}, z^{(n+1)} = \text{Project}(x^{(\nu)}, z^{(\nu)}, \tilde{L}^{(\nu+1)})$
7:         $L^{(\nu+1)} = \text{Forget}(z^{(\nu+1)}, \tilde{L}^{(\nu+1)})$
8:     **end while**
9:     **return** $x$
10: **end function**

---

## E.2 PROJECT AND FORGET FOR SUMS OF SUPERMODULAR CONES

We adapt PROJECT AND FORGET for optimizing over the cone of $\pi$-supermodular functions. The algorithm begins by initializing $L^{(\nu)}$, for $\nu = 0$, as the empty list. This will keep track of the violated and active constraints. The first step in PROJECT AND FORGET is to implement an efficient oracle $\mathcal{Q}$ that, given a query point $x^{(\nu)}$, returns a list $L$ of violated constraints. This $L$ is merged $L^{(\nu)}$ to get $L^{(\nu+1)}$. All violated constraints need not be returned, but each constraint violated by $x^{(\mu)}$ should be returned with positive probability. Here we detail two options.

---

**Algorithm 4** Project and Forget Algorithms.

---

1: **function** PROJECT($x, z, L$)
2:  **for** $H_i = \{y : \langle a_i, y \rangle = b_i\} \in L$ **do**
3:   Find $x^*, \theta$ by solving $\nabla f(x^*) - \nabla f(x) = \theta a_i$ and $x^* \in H_i$
4:   $c_i = \min(z_i, \theta)$
5:   $x \leftarrow x_{new}$
6:   $x_{new} \leftarrow$ such that $\nabla f(x_{new}) - \nabla f(x) = c_i a_i$
7:   $z_i \leftarrow z_i - c_i$
8:  **end for**
9:  **return** $x, z$
10: **end function**
11: **function** FORGET($z, L$)
12:  **for** $H_i = \{x : \langle a_i, x \rangle = b_i\} \in L$ **do**
13:   **if** $z_i == 0$ **then**
14:    Forget $H_i$
15:   **end if**
16:  **end for**
17:  **return** $L$
18: **end function**

---

- **Deterministic Oracle.** Go through all $\binom{n}{2} 2^{n-2}$ constraints and see which are violated. There are exponentially many such constraints.

- **Random Oracle** For each of the $\binom{n}{2}$ pair $i \neq j$, we sample $5n$ of the $2^{n-2}$ constraints. We return the violated ones. Each violated constraint has a positive probability of being returned.

Our objective function $\|f - g\|_{\ell_2}^2$ is quadratic, so for the project step, we use the formula for a quadratic objective from Sonthalia and Gilbert (2022). Specifically, we iteratively project $x^{(\nu)}$ onto each constraint in $L^{(\nu+1)}$ and update $z^{(\nu)}$ to get $x^{(\nu+1)}$ and $z^{(\nu+1)}$. There is nothing to adapt in the forget step: we remove from $L^{(\nu+1)}$, inactive constraints with $z_i^{(\nu+1)} = 0$.

**Remark 54.** With these adaptations Theorem 53 applies. Hence we have a linear rate of convergence. We take an exponential amount of time per iteration with the deterministic oracle. However, with the random oracle, we may take polynomial time per iteration. We might still need exponential time per iteration if there are exponentially many active constraints. That is, $L^{(\nu+1)}$ becomes exponentially long. See experiment in Figure 2c and Appendix G for running times for computing low-rank approximations.

## F  DETAILS ON THE SET FUNCTION OPTIMIZATION GUARANTEES

We provide details and proofs for the results in Section 4 and discuss related prior work.

### F.1  PREVIOUS RESULTS ON MAXIMIZATION OF MONOTONE SET FUNCTIONS

We present the results from prior work that are the basis of our comparison.

**Submodular functions.**  We begin with the following classical result.

**Theorem 55** (Nemhauser et al. (1978))**.** *Fix a normalized monotone submodular function $f : 2^V \to \mathbb{R}$ and let $\{S_i\}_{i \geq 0}$ be the greedily selected sets for constrained cardinality problem. Then for all positive integers $m$ and $\ell$,*

$$f(S_\ell) \geq \left(1 - e^{-\ell/m}\right) \max_{\Omega : |\Omega| \leq m} f(\Omega).$$

*In particular, for $\ell = m$, $f(S_m)$ is a $1 - e^{-1}$ approximation for the optimal solution.*

Călinescu et al. (2011); Filmus and Ward (2012) extended the above result to the matroid constraint problem and obtained the following.

**Theorem 56** (Călinescu et al. (2011)). *There is a randomized algorithm that gives a $(1 - e^{-1})$-approximation (in expectation over the randomization in the algorithm) to the problem of maximizing a monotone, non-negative, submodular function $f : 2^{[n]} \to \mathbb{R}$ subject to matroid constraint $\mathcal{M}$ given by a membership oracle. The algorithm runs in $\tilde{O}(n^8)$ time.*

**Theorem 57** (Filmus and Ward (2012)). *Let $f$ be a normalized, positive, monotone, submodular function, and let $\mathcal{M}$ be a rank $\rho$ matroid. For all $\epsilon > 0$, there exists a randomized algorithm that is a $1 - e^{-1} - \epsilon$ approximation for the maximization of $f$ over $\mathcal{M}$ that queries $f$ at most $O(\epsilon^{-1}\rho^2 n \log(n))$ times.*

**Total curvature.** The total curvature $\hat{\alpha}$ measures how far a submodular function is from being modular, see Definition 23). It can be used to refine Theorem 55. We have $\hat{\alpha} = 0$ if and only if the function is modular, and that $\hat{\alpha} \leq 1$ for any submodular function. Then Conforti and Cornuéjols (1984) prove the following.

**Theorem 58** (Conforti and Cornuéjols (1984, Theorem 2.3)). *If $\mathcal{M}$ is a matroid and $f$ is a normalized, monotone submodular function with total curvature $\hat{\alpha}$, then* GREEDY *returns a set $S$ with*

$$f(S) \geq \frac{1}{1 + \hat{\alpha}} \max_{\Omega \in \mathcal{M}} f(\Omega).$$

Building on this, Sviridenko et al. (2017) present a the NON-OBLIVIOUS LOCAL SEARCH GREEDY algorithm.

**Theorem 59** (Sviridenko et al. (2017, Theorem 6.1)). *For every $\epsilon > 0$, matroid $\mathcal{M}$, and monotone, non-negative submodular function $f$ with total curvature $\hat{\alpha}$,* NON OBLIVOUS LOCAL SEARCH GREEDY *produces a set $S$ with high probability, in $O(\epsilon^{-1}poly(n))$ time that satisfies*

$$f(S) \geq (1 - \hat{\alpha}e^{-1} + O(\epsilon)) \max_{\Omega \in \mathcal{M}} f(\Omega).$$

Further Sviridenko et al. (2017), provide the following result to show that no polynomial time algorithm can do better.

**Theorem 60** (Sviridenko et al. (2017)). *For any constant $\delta > 0$ and $c \in (0, 1)$, there is no $(1 - ce^{-1} + \delta)$ approximation algorithm for the cardinality constraint maximization problem for monotone submodular functions $f$ with total curvature $\hat{a}^f \leq c$, that evaluates $f$ on only a polynomial number of sets.*

For arbitrary monotone increasing functions, they define a curvature $c$ (different from Definition 71), which agrees with $\hat{\alpha}$ (Definition 23) for monotone submodular functions, to get a $(1 - c)$ approximation ratio with the GREEDY algorithm.

**Approximately submodular functions.** Recall the submodularity ratio $\gamma$ and the generalized curvature $\alpha$ from Definition 24.

**Theorem 61** (Bian et al. (2017)). *Fix a non-negative monotone function $f : 2^V \to \mathbb{R}$ with submodularity ratio $\gamma$ and curvature $\alpha$. Let $\{S_i\}_{i \geq 0}$ be the sequence produced by the* GREEDY *algorithm. Then for all positive integers $m$,*

$$f(S_m) \geq \frac{1}{\alpha}\left[1 - \left(\frac{m - \alpha\gamma}{m}\right)^m\right] \max_{\Omega:|\Omega| \leq m} f(\Omega)$$

$$\geq \frac{1}{\alpha}(1 - e^{-\alpha\gamma}) \max_{\Omega:|\Omega| \leq m} f(\Omega).$$

*Further, the above bound is tight for the* GREEDY *algorithm.*

Departing from cardinality constrained matroids, Buchbinder et al. (2014); Chen et al. (2018) optimize non-negative monotone set functions with submodularity ratio $\gamma$ over general matroids.

**Theorem 62** (Chen et al. (2018)). *The* RESIDUAL RANDOM GREEDY *algorithm has an approximation ratio of at least $(1 + \gamma^{-1})^{-2}$ for the problem of maximizing a non-negative monotone set function with submodularity ratio $\gamma$ subject to a matroid constraint.*

Following this, Gatmiry and Gomez-Rodriguez (2018) looked at the approximation for the GREEDY Algorithm for set functions with submodularity ratio $\gamma$ and curvature $\alpha$, subject to general matroid constraints.

**Theorem 63** (Gatmiry and Gomez-Rodriguez (2018)). *Given a matroid $\mathcal{M}$ with rank $\rho \geq 3$ and a monotone set function $f$ with submodularity ratio $\gamma$ the* GREEDY *algorithm returns a set $S$ such that*

$$f(S) \geq \frac{0.4\gamma^2}{\sqrt{\rho\gamma} + 1} \max_{\Omega \in \mathcal{M}} f(\Omega)$$

**Theorem 64** (Gatmiry and Gomez-Rodriguez (2018)). *Given a matroid $\mathcal{M}$ and a monotone set function $f$ with curvature $\alpha$ the* GREEDY *algorithm returns a set $S$ such that*

$$f(S) \geq \left(1 + \frac{1}{1-\alpha}\right)^{-1} \max_{\Omega \in \mathcal{M}} f(\Omega).$$

**Other measures of approximate submodularity.** Krause et al. (2008) say a function is $\epsilon$ submodular if for all $A \subset B \subset V$, we have that $\Delta(e|A) \geq \Delta(e|B) - \epsilon$. In this case, they proved that with cardinality constraint, GREEDY returns a set $S_m$ such that $f(S_m) \geq (1 - e^{-1}) \max_{\Omega:|\Omega| \leq m} f(\Omega) - m\epsilon$. Du et al. (2008) study the problem when $f$ is submodular over certain collections of subsets of $V$. Here they provide an approximation result for a greedy algorithm used to solve

$$\min c(S), \quad \text{subject to: } f(S) \geq C, S \subseteq V,$$

where $c(S)$ is a non-negative modular function, and $C$ is a constant. Finally, Horel and Singer (2016) look at set functions $f$, such that there is a submodular function $g$ with $(1 - \epsilon)g(S) \leq f(S) \leq (1 + \epsilon)g(S)$ for all $S \subset V$. They provide results on the sample complexity for querying $f$ as a function of the error level for the cardinality constraint problem.

## F.2 MAXIMIZATION OF MONOTONE FUNCTIONS WITH BOUNDED ELEMENTARY SUBMODULAR RANK

We study functions with low elementary submodular rank. Recall our Definition 26:

**Definition 26.** Given $A \subseteq B \subseteq V$, define $\Pi(A, B) := \{C \subseteq V : C \cap B = A\}$. Given a set function $f : 2^V \to \mathbb{R}$, we let $f_{A,B}$ denote its restriction to $\Pi(A, B)$.

The sets that contain $i$ are $\Pi(\{i\}, \{i\})$ and the sets that do not contain $i$ are $\Pi(\emptyset, \{i\})$.

**Proposition 65.** *Let $f_i$ be an $\{i\}$-submodular function. Then $f_{\{i\},\{i\}}$ and $f_{\emptyset,\{i\}}$ are submodular.*

*Proof.* For all $S_1, S_2 \in \Pi(\{i\}, \{i\})$, we know that $i \in S_1, S_2$. Hence the linear ordering on the $i^{th}$ coordinate does affect the computation of the least upper bound and greatest lower bound. Thus, $S_1 \wedge S_2 = S_1 \cap S_2$ and $S_1 \vee S_2 = S_1 \cup S_2$. Hence the submodularity inequalities from Definition 1 hold. Similarly, for $S_1, S_2 \in \Pi(\emptyset, \{i\})$, we have $i \notin S_1, S_2$. Hence $S_1 \wedge S_2 = S_1 \cap S_2$ and $S_1 \vee S_2 = S_1 \cup S_2$. $\square$

With this, we can now prove our Proposition 27:

**Proposition 27.** *A set function $f$ has elementary submodular rank $r + 1$, with decomposition $f = f_0 + f_{i_1} + \cdots + f_{i_r}$, if and only if $f_{A,B}$ is submodular for $B = \{i_1, \ldots, i_r\}$ and any $A \subseteq B$.*

*Proof.* If $f$ has such a decomposition, then the fact that the pieces $f_{A,B}$ are submodular follows from Proposition 65, using that a sum of $(1, \ldots, 1)$-submodular functions is $(1, \ldots, 1)$-submodular. For the converse, assume that $f_{A,B}$ is submodular for (without loss of generality) $B = \{1, \ldots, r\}$ and any $A \subseteq B$. Then $f \in \mathcal{L}_\xi$, where $\xi_{ij} = -1$ for all $i \neq j$ with $i, j \geq r + 1$. The cone $\mathcal{L}_\xi$ is a sum of $-\mathcal{L}_{(1,\ldots,1)}$ and the $\{i\}$-submodular cones for all $i \leq r$. $\square$

We can now prove Proposition 28:

**Proposition 28.** *If $f$ is a set function with submodularity ratio $\gamma$ and generalized curvature $\alpha$, then $f_{A,B}$ has submodularity ratio $\gamma_{A,B} \geq \gamma$ and generalized curvature $\alpha_{A,B} \leq \alpha$.*

*Proof of Proposition 28.* This follows from Definitions 24 and 29, as in restriction $f_{A,B}$ we have fewer sets $S, T$, in the definition, for which in the inequality in the definitions need to hold. $\square$

We are now ready to prove our Theorem 30:

**Theorem 30** (Guarantees for R-SPLIT). *Let $\mathcal{A}$ be an algorithm for matroid constrained maximization of set functions, such that for any monotone, non-negative function $g\colon 2^W \to \mathbb{R}$, $|W| = m$, with generalized curvature $\alpha$ and submodularity ratio $\gamma$, algorithm $\mathcal{A}$ makes $O(q(m))$ queries to the value of $g$, where $q$ is a polynomial, and returns a solution with approximation ratio $R(\alpha, \gamma)$. If $f\colon 2^{[n]} \to \mathbb{R}$ is a monotone, non-negative function with generalized curvature $\alpha$, submodularity ratio $\gamma$, and elementary submodular rank $r + 1$, then R-SPLIT with subroutine $\mathcal{A}$ runs in time $O(2^r n^r q(n))$ and returns a solution with approximation ratio $\max\{R(\alpha, \gamma), R(\alpha_r, 1)\}$.*

*Proof of Theorem 30.* We first discuss the computational cost. If we run $\mathcal{A}$ on $f$, we get an approximation ratio of at least $R(\alpha, \gamma)$, at a cost of $O(q(n))$. For R-SPLIT, there are $\binom{n}{r} = O(n^r)$ sets $B \subset V$ of cardinality $r$. For each, there are $2^r$ subsets $A \subseteq B$. Hence we have $2^r \binom{n}{r}$ possible $\Pi(A, B)$. On each, we run $\mathcal{A}$, which runs in $O(q(n - |A|))$, because $f_{A,B}$ can be regarded as a function on $2^{V \setminus B}$. The final step is to pick the optimal value among the solutions returned for each subproblem, which can be done in $O(2^r \binom{n}{r})$ time. Hence the overall cost is $O(q(n)) + 2^r \binom{n}{r} O(q(n - |A|)) + O(2^r \binom{n}{r}) = O(2^r n^r q(n))$.

Now we discuss the approximation ratio. The solution returned for $f_{A,B}$ has approximation ratio $R(\alpha_{A,B}, \gamma_{A,B})$, by the assumed properties of $\mathcal{A}$. Since $f$ has elementary submodular rank $r+1$, there exist submodular $f_0$ and elementary submodular $f_{i_1}, \ldots, f_{i_r}$ such that $f = f_0 + f_{i_1} + \cdots + f_{i_r}$. Let $B_f = \{i_1, \ldots, i_r\}$. Then, for this set $B_f$ and any $A \subseteq B_f$, we know that $f_{A,B_f}$ is submodular, by Proposition 27, and hence $\gamma_{A,B_f} = 1$. Picking the optimum value among the solutions returned for the subproblems involving $B_f$ ensures an overall approximation ratio with $\min_{A \subseteq B_f} \gamma(A, B_f) = 1$, that is, $R(\alpha_r, 1)$. In summary, we are guaranteed to obtain a final solution with approximation ratio $\max\{R(\alpha, \gamma), R(\alpha_r, 1)\}$. □

**Remark 66.** With knowledge of the elementary cones involved in the decomposition of $f$, the set $B_f$ from the proof of Theorem 30, we only need to consider the subproblems that involve $B_f$. This gives $2^r$ subproblems instead of $2^r \binom{n}{r}$.

Let us now instantiate corollaries of Theorem 30. We fix the elementary rank to be $r + 1$. The runtime will be exponential in $r$ but polynomial in $n$.

**Corollary 67.** *If $f$ has elementary submodular rank $r + 1$, then for the cardinality constrained problem, for all $\epsilon > 0$, the approximation ratio for the R SPLIT GREEDY is $\max(R(\alpha, \gamma), (1 - \hat{\alpha}e^{-1}) - O(\epsilon))$. The algorithm runs in $O(\epsilon^{-1} 2^r n^r \cdot poly(n))$.*

*Proof.* Use NON-OBLIVIOUS LOCAL SEARCH GREEDY as the subroutine and Theorem 59 for the guarantee. □

**Corollary 68.** *If $f$ is a non-negative monotone function with submodularity ratio $\gamma$, curvature $\alpha$, and elementary submodular rank $r + 1$, then for the cardinality constrained problem, the R SPLIT GREEDY algorithm has an approximation ratio of $\alpha_r^{-1}(1 - e^{-\alpha_r})$ and runs in $O(2^r n^r \cdot nm)$ time.*

*Proof.* Use GREEDY as the subroutine and Theorem 61 for the guarantee. □

**Corollary 69.** *For a non-negative monotone function $f$ and elementary submodular rank $r + 1$ for the matroid constrained problem the R SPLIT GREEDY has an approximation ratio of $(1 - e^{-1})$ and runs in $\tilde{O}(2^r n^r \cdot n^8)$ time.*

*Proof.* Use the algorithm from Călinescu et al. (2011) and Theorem 56 for the guarantee. It remains to show that the procedure from Călinescu et al. (2011) terminates in $\tilde{O}(n^8)$, even if the input is not submodular. The algorithm from Călinescu et al. (2011) consists of two steps. First is the CONTINUOUS GREEDY algorithm. Second, is PIPAGE ROUND (Ageev and Sviridenko (2004)). From Călinescu et al. (2011) we have that CONTINUOUS GREEDY terminates after a fixed number of steps. This would be true even if the input function $f$ is not submodular and always returns a point in the base polytope of the matroid. Second, using (Călinescu et al., 2011, Lemma 3.5), we see that PIPAGE ROUND, terminates in polynomial time for any point in the base matroid of the polytope. □

In Table 2 we summarize the results from Corollaries 67, 68, and 69 and how they compare with previous results given above in Theorems 56, 59, 61, 62, 63, and 64.

**Theorem 33** (Lower Bound). *Any deterministic procedure that achieves an $O(1)$ approximation ratio for maximizing elementary submodular rank-$(r+1)$ set functions requires at least $2^r$ function queries.*

*Proof.* Assume for contradiction that such a procedure $\mathcal{A}$ exists. Let $f$ be a set function with elementary rank-$(r+1)$ and let $f_1, \ldots, f_{2^r}$ be the submodular splits of the functions. Since the procedure samples fewer than $2^r$ values. There is a piece $f_i$ that is not queried. Consider a modified function $\hat{f}$, with the same piece, i.e., $\hat{f}_j = f_j$ except that $\hat{f}_i = f_i + c$. Since $c$ can be arbitrary, and the procedure is deterministic, it will not query $\hat{f}_i$. Hence will not return an $O(1)$ approximation. Hence contradiction. $\square$

### F.3 Minimization of Ratios of Set Functions

The second application of our decomposition is to minimize ratios of set functions. We give definitions and previous results, with details in Appendix F.

**Problem 70.** *Given set functions $f$ and $g$, we seek $\min_S \frac{f(S)}{g(S)}$.*

Bai et al. (2016) obtain guarantees for RATIO GREEDY, see Algorithm 5, for different submodular or modular combinations of $f$ and $g$. To quantify the approximation ratio for non-sub-/modular $f, g$, we need a few definitions.

---

**Algorithm 5** RATIO GREEDY

1: **function** RATIO GREEDY($f,g$)
2:    Initialize: $S_0 = \emptyset$, $R = V$
3:    **while** $R \neq \emptyset$ **do**
4:      $u = \arg\min_{v \in R} \frac{f(\{v\} \cup S_i)}{g(\{v\} \cup S_i)}$
5:      $S_{i+1} = S_i \cup \{u\}$
6:      $R = \{v \in R : g(\{v\} \cup S_{i+1}) > 0\}$
7:    **end while**
8:    **return** $S_i$
9: **end function**

---

**Definition 71** (Bogunovic et al. (2018)). • The *generalized inverse curvature* of a non-negative set function $f$ is the smallest $\tilde{\alpha}^f$ such that for all $T, S \in 2^V$ and for all $e \in S \setminus T$,

$$\Delta(e|S \setminus \{e\}) \geq (1 - \tilde{\alpha}^f)\Delta(e|(S \setminus \{e\}) \cup T).$$

• The *curvature* of $f$ with respect to $X$ is $\hat{c}^f(X) := 1 - \frac{\sum_{e \in X}(f(X) - f(X \setminus \{e\}))}{\sum_{e \in X} f(\{e\})}$.

It is known that $f$ is submodular if and only if $\tilde{\alpha}^f = 0$. Using these notions, Qian et al. (2017) obtain an approximation guarantee for minimization of $f/g$ with monotone submodular $f$ and monotone $g$, and Wang et al. (2019) obtain a guarantee when both $f$ and $g$ are monotone.

**Functions with bounded elementary rank.** We formulate results for R-SPLIT with GREEDY RATIO subroutine, shown in Algorithm 6. We improve on previous guarantees if the functions have low elementary submodular rank. First, we split only $f$.

**Theorem 72** (Guarantees for R-SPLIT GREEDY RATIO I). *For the minimization of $f/g$ where $f, g$ are normalized positive monotone functions, assume $f$ has elementary submodular rank $r + 1$. Let $X^*$ be the op-*

---

**Algorithm 6** R-SPLIT RATIO

1: **function** R-SPLIT RATIO($f, g, r, \mathcal{A}$ - GREEDY RATIO)
2:    **for** $A \subseteq B \subseteq V$ with $|B| = r$ **do**
3:      run $\bar{\mathcal{A}}$ on $\bar{f}_{A,B}$ and $g_{A,B}$
4:    **end for**
5:    run $\mathcal{A}$ on $f, g$
6:    **return** Best seen set
7: **end function**

---

*timal solution. Then R-SPLIT with GREEDY RATIO subroutine at a time complexity of $O(2^r n^r \cdot n^2)$, has approximation ratio*

$$\frac{1}{\gamma_{\emptyset,|X^*|}^g} \frac{|X^*|}{1 + (|X^*| - 1)(1 - \hat{c}^f(X^*))}.$$

*If, in addition, $g$ is submodular then the approximation ratio is $1/(1 - e^{\hat{\alpha}^f - 1})$.*

The first statement provides the same guarantee as a result of Qian et al. (2017) but for a more general class of functions. Our result can be interpreted as grading the conditions of Wang et al. (2019) to obtain similar guarantees as the more restrictive results of Qian et al. (2017). Next, we split both $f$ and $g$.

**Theorem 73** (Guarantees for R-SPLIT GREEDY RATIO II). *Assume $f$ and $g$ are normalized positive monotone functions, with elementary submodular ranks $r_f + 1$ and $r_g + 1$. For the minimization of $f/g$, the algorithm R-SPLIT with GREEDY RATIO subroutine at a time complexity of $O(2^{r_f + r_g} n^{r_f + r_g} \cdot n^2)$, has approximation ratio $\frac{1}{1 - e^{\hat{\alpha}^f - 1}}$,*

Table 3: Approximation ratios and time complexity for minimizing $f/g$ for monotone, normalized functions $f$ and $g$ on $2^{[n]}$. Here $f$ has total curvature $\hat{\alpha}^f$, generalized curvature $\alpha^f$, generalized inverse curvature $\tilde{\alpha}^f$, curvature $\hat{c}^f$, and elementary submodular rank $r_f + 1$. $g$ has submodularity ratio $\gamma^g$ and elementary submodular rank $r_g + 1$. $X^*$ is the optimal solution. Prior works are discussed in detail in Appendix F.

|  | Numerator | Denominator | Approx. Ratio | Time | Ref. |
|---|---|---|---|---|---|
| **Prior Work** | Modular | Modular | $1$ | $O(n^2)$ | Bai et al. (2016) |
| | Modular | Submodular | $1 - e^{-1}$ | $O(n^2)$ | Bai et al. (2016) |
| | Submodular | Submodular | $\frac{1}{1 - e^{\hat{\alpha}^f - 1}}$ | $O(n^2)$ | Bai et al. (2016) |
| | Submodular | - | $\frac{1}{\gamma^g_{\emptyset,|X^*|}} \frac{|X^*|}{1 + (|X^*| - 1)(1 - \hat{c}^f(X^*))}$ | $O(n^2)$ | Qian et al. (2017) |
| | - | - | $\frac{1}{\gamma^g_{\emptyset,|X^*|}} \frac{|X^*|}{1 + (|X^*| - 1)(1 - \alpha^f)(1 - \tilde{\alpha}^f)}$ | $O(n^2)$ | Wang et al. (2019) |
| **This Work** | Low Rank | - | $\frac{1}{\gamma^g_{\emptyset,|X^*|}} \frac{|X^*|}{1 + (|X^*| - 1)(1 - \hat{c}^f(X^*))}$ | $O(2^{r_f} n^{r_f} \cdot n^2)$ | Thm 72 |
| | Low Rank | Submodular | $\frac{1}{1 - e^{\hat{\alpha}^f - 1}}$ | $O(2^{r_f} n^{r_f} \cdot n^2)$ | Thm 72 |
| | Low Rank | Low Rank | $\frac{1}{1 - e^{\hat{\alpha}^f - 1}}$ | $O(2^{r_f + r_g} n^{r_f + r_g} \cdot n^2)$ | Thm 73 |

This improves the approximation ratio guarantee of Wang et al. (2019) to that of Bai et al. (2016), which was valid only if $f, g$ are both submodular. Table 3 compares our results with prior work.

### F.4 PRIOR RESULTS

For Problem 70 of minimizing ratios of set functions, RATIO GREEDY from Algorithm 5 has the following guarantees.

**Theorem 74** (Bai et al. (2016)). *For the ratio of set function minimization problem* $\min f/g$, GREEDY RATIO *has the following approximation ratios:*

1. *If $f, g$ are modular, then it finds the optimal solution.*

2. *If $f$ is modular and $g$ is submodular, then it finds a $1 - e^{-1}$ approximate solution.*

3. *If $f$ and $g$ are submodular, then it finds a $1/(1 - e^{\hat{\alpha}^f - 1})$ approximate solution, where $\hat{\alpha}^f$ is the total curvature of $f$.*

To quantify the approximation ratio of the GREEDY RATIO algorithm, we recall the definitions of generalized inverse curvature (Definition 71) as well as alternative notions of the submodularity ratio (Definition 24) and curvature (Definition 71).

Main results that give guarantees for the ratio of submodular function minimization are as follows.

**Theorem 75** (Qian et al. (2017)). *For minimizing the ratio $f/g$ where $f$ is a positive monotone submodular function and $g$ is a positive monotone function,* GREEDY RATIO *finds a subset $X \subseteq V$ with*

$$\frac{f(X)}{g(X)} \leq \frac{1}{\gamma^g_{\emptyset,|X^*|}} \frac{|X^*|}{1 + (|X^*| - 1)(1 - \hat{c}^f(X^*))} \frac{f(X^*)}{g(X^*)},$$

*where $X^*$ is the optimal solution and $\gamma^g$ is the submodularity ratio of $g$.*

**Theorem 76** (Wang et al. (2019)). *For minimizing the ratio $f/g$ where $f$ and $g$ are normalized non-negative monotone set functions,* GREEDY RATIO *outputs a subset $X \subseteq V$, such that*

$$\frac{f(X)}{g(X)} \leq \frac{1}{\gamma^g_{\emptyset,|X^*|}} \frac{|X^*|}{1 + (|X^*| - 1)(1 - \alpha^f)(1 - \tilde{\alpha}^f)} \frac{f(X^*)}{g(X^*)},$$

*where $X^*$ is the optimal solution, $\gamma^g$ is the submodularity ratio of $g$, and $\alpha^f$ (resp. $\tilde{\alpha}^f$) are the generalized curvature (resp. generalized inverse curvature) of $f$.*

### F.5 Minimization of Ratios of Set Functions with Bounded Elementary Rank

We now formulate results for our R-SPLIT with GREEDY RATIO subroutine.

**Theorem 72** (Guarantees for R-SPLIT GREEDY RATIO I). *For the minimization of $f/g$ where $f, g$ are normalized positive monotone functions, assume $f$ has elementary submodular rank $r + 1$. Let $X^*$ be the optimal solution. Then R-SPLIT with GREEDY RATIO subroutine at a time complexity of $O(2^r n^r \cdot n^2)$, has approximation ratio*

$$\frac{1}{\gamma^g_{\emptyset,|X^*|}} \frac{|X^*|}{1 + (|X^*| - 1)(1 - \hat{c}^f(X^*))}.$$

*If, in addition, $g$ is submodular then the approximation ratio is $1/(1 - e^{\hat{\alpha}^f - 1})$.*

*Proof of Theorem 72.* The statement follows from analogous arguments to those in the proof of Theorem 30. There is a set $B_f$ such that $f_{A,B_f}$ is submodular on all $A \subseteq B_f$. We use Theorem 75 to get the approximation ratio.

If $g$ is submodular, then its restrictions are submodular, and we minimize the ratio of two submodular functions. Hence can use Theorem 74 to obtain the approximation ratio. □

Next, we consider our Theorem 73 splitting both the numerator $f$ and the denominator $g$:

**Theorem 73** (Guarantees for R-SPLIT GREEDY RATIO II). *Assume $f$ and $g$ are normalized positive monotone functions, with elementary submodular ranks $r_f + 1$ and $r_g + 1$. For the minimization of $f/g$, the algorithm R-SPLIT with GREEDY RATIO subroutine at a time complexity of $O(2^{r_f + r_g} n^{r_f + r_g} \cdot n^2)$, has approximation ratio $\frac{1}{1 - e^{\hat{\alpha}^f - 1}}$,*

*Proof of Theorem 73.* Use $r = r_f + r_g$ in Algorithm 6. Then there is a $B_f$ such that $f_{A,B_f}$ is submodular and a $B_g$ such that $g_{A,B_g}$ is submodular. Let $B_{f,g} = B_f \cup B_g$. Then $|B_{f,g}| \leq |B_f| + |B_g| = r_f + r_g$ and both $f_{A,B_{f,g}}$ and $g_{A,B_{f,g}}$ are submodular. We minimize a ratio of submodular functions, and use Theorem 74 for the approximation ratio. □

**Remark 77.** We have assumed that we do not know the decomposition of $f$ into elementary submodular functions. However, if we knew the decomposition, then we can extend any optimization for submodular functions to elementary submodular rank-$r$ functions, incurring a penalty of $2^r$. This gives another approach for set function optimization: first compute a low-rank approximation and then run our procedure on this low-rank approximation. We describe an algorithmic implementation of this in Appendix E.

### F.6 Comparison of Elementary Submodular Rank and Curvature Notions

As mentioned in Remark 25, for monotone increasing $f$, we have $\gamma \in [0, 1]$, with $\gamma = 1$ iff $f$ is submodular. Values less than one correspond to violations of the diminishing returns property of submodularity. Moreover, for a monotone increasing function $f$, we have $\alpha \in [0, 1]$ and $\alpha = 0$ if and only if $f$ is supermodular. Note this latter is a condition that the function is supermodular and not that the function is submodular. Thus, if $\alpha = 0$ and $\gamma = 1$, then $f$ is *both* supermodular and submodular, and thus it is modular.

## G Details on the Experiments

### G.1 Types of Functions

We consider four types of objective functions. The first three are commonly encountered in applications. The fourth are random monotone functions.

**Determinantal functions.** Let $\Sigma = XX^T$ be a positive definite matrix, where $X \in \mathbb{R}^{n \times d}$ is a Gaussian random matrix (i.e., entries are i.i.d. samples from a standard Gaussian distribution). To ensure $\Sigma$ is positive definite, we impose $d \geq n$. Given $S \subseteq [n]$, denote by $\Sigma_S$ the $|S| \times |S|$ matrix indexed by the elements in $S$. Given $\sigma \in \mathbb{R}$, we define

$$f(S) := \det(I + \sigma^{-2}\Sigma_S), \quad S \subseteq [n].$$

Bian et al. (2017, Proposition 2) show that $f$ is supermodular. This type of functions appear in determinantal point processes, see Kulesza and Taskar (2012).

**Bayesian A-optimality functions.** The Bayesian A-optimality criterion in experimental design seeks to minimize the variance of a posterior distribution as a function of the set of observations. Let $x_1, \ldots, x_n \in \mathbb{R}^d$ be $n$ data points. For any $S \subseteq [n]$, let $X_S \in \mathbb{R}^{n \times |S|}$ be the matrix collecting the data points with index in $S$. Let $\theta \sim \mathcal{N}(0, \beta^{-2}I)$ be a parameter and let $y_S = \theta^T X_S + \xi$, where $\xi \sim \mathcal{N}(0, \sigma^2 I)$. Let $\Sigma_{\theta|y_S}$ be the posterior covariance of $\theta$ given $y_S$. Then we define

$$f(S) = \text{Tr}(\beta^{-2}I) - \text{Tr}(\Sigma_{\theta|y_S}) = \frac{d}{\beta^2} - \frac{1}{\beta^2}\text{Tr}((I + (\beta\sigma)^{-2}X_S X_S^T)^{-1}).$$

Maximizing $f$ identifies a set of observations that minimizes the variance of the posterior. Bian et al. (2017) provide bounds on $\alpha$ and $\gamma$ for this function.

**Column subset selection.** Given a matrix $A$, we ask for a subset $S$ of the columns that minimizes

$$f(S) = \|A\|_F^2 - \|A_S A_S^\dagger A\|_F^2.$$

Here, $A_S^\dagger$ is the Moore-Penrose pseudoinverse. Hence $A_S A_S^\dagger$ is the orthogonal projection matrix onto the column space of $A_S$.

**Random functions.** We take a uniform random sample from $[0, 1]$ of size $2^n$ and sort it in increasing order as a list $L$. We then construct a monotone function $f$ by assigning to $f(\emptyset)$ the smallest value in $L$ and then, for $M = 1, \ldots, n$, assigning to $f(S)$, $S \subseteq [n]$, $|S| = M$, in any order, the next $\binom{n}{M}$ smallest elements of $L$.

### G.2 SUBMODULARITY RATIO AND GENERALIZED CURVATURE

Here we let $n = 8$. We sampled five different sample functions for each function type and computed $\alpha_r$ and $\gamma_r$ for $r = 0, 1, 2, 3, 4$, where $\alpha_0$ and $\gamma_0$ are by convention the generalized curvature and submodularity ratio for the original function.

**Determinantal.** Here we first sample $X \in \mathbb{R}^{n \times n}$ with i.i.d. standard Gaussian entries. We then form $\Sigma = XX^T$. We also set $\sigma = 0.1$.

**Bayesian A-optimality.** Here we first sample $X \in \mathbb{R}^{60 \times n}$ with i.i.d. standard Gaussian entries. We use $\beta = 0.1$ and $\sigma = 0.1$

**Column subset.** Here we first sample $A \in \mathbb{R}^{20 \times n}$ with i.i.d. standard Gaussian entries.

**Random.** There are no hyperparameters to set.

Figure 8 shows $\alpha_r$ and $\gamma_r$.

### G.3 LOW ELEMENTARY RANK APPROXIMATIONS

Here we let $n = 7$. We sampled 50 different sample functions for each function and then computed the low elementary rank approximation for $r + 1 = 1, 2, 3, 4, 5, 6, 7$. Appendix E.1 discusses the details of the algorithm used to compute the low-rank approximations.

**Determinantal.** Here we first sample $X \in \mathbb{R}^{n \times 2n}$ with i.i.d. standard Gaussian entries. We then form $\Sigma = XX^T$. We also set $\sigma = 0.1$.

**Bayesian A-optimality.** Here we first sample $X \in \mathbb{R}^{60 \times n}$ with i.i.d. standard Gaussian entries. We use $\beta = 1$ and $\sigma = 0.01$

**Column subset.** Here we first sample $A \in \mathbb{R}^{60 \times n}$ with i.i.d. standard Gaussian entries.

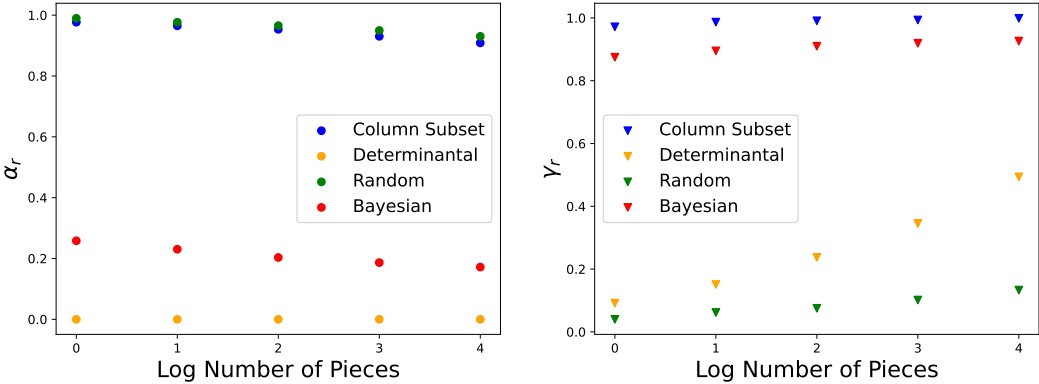

Figure 8: Shown are (a) $\alpha_r$ and (b) $\gamma_r$ for the four different function types for splitting into 1,2,4,8, and 16 pieces.

**Random.** There are no hyperparameters to set.

### G.4 R-SPLIT GREEDY WITH SMALL $n$

Here we let $n = 20$. We sampled 50 different sample functions for each function and for each function, ran GREEDY and R-SPLIT with GREEDY as the subroutine and $r = 1, 2, 3$.

**Determinantal.** Here we sample $X \in \mathbb{R}^{n \times 2n}$ with i.i.d. standard Gaussian entries. We then form $\Sigma = XX^T$. We also set $\sigma = 0.1$.

**Bayesian A-optimality.** Here we first sample $X \in \mathbb{R}^{60 \times n}$ with i.i.d. standard Gaussian entries. We use $\beta = 1$ and $\sigma = 0.01$

**Column subset.** Here we sample $A \in \mathbb{R}^{40 \times n}$ with i.i.d. standard Gaussian entries.

**Random.** There are no hyperparameters to set.

### G.5 R-SPLIT GREEDY WITH LARGE $n$

Here we let $n = 25, 50, 75, 100, 150, 200, 250, 300, 350, 400, 450, 500$. We sampled five different sample functions for each function and, for each function, ran GREEDY and R-SPLIT with GREEDY as the subroutine and $r = 1$.

**Determinantal.** Here we first sample $X \in \mathbb{R}^{n \times n}$ with i.i.d. standard Gaussian entries. We then form $\Sigma = XX^T$. We also set $\sigma = 1$.

**Bayesian A-optimality.** Here we first sample $X \in \mathbb{R}^{60 \times n}$ with i.i.d. standard Gaussian entries. We use $\beta = 0.1$ and $\sigma = 0.1$

**Column subset.** We use MNIST dataset for this problem.

### G.6 INITIAL SEED GREEDY

To compare algorithms with the same time dependence on $n$, we compare against a method we term INITIAL SEED GREEDY. This algorithm provides an initial set to GREEDY. That is, instead of starting at the empty set, it starts at a provided set. We then compare INITIAL SEED GREEDY by providing all $\binom{n}{r}$ seeds and compare to R-SPLIT GREEDY.

For $r = 1$ and the DETERMINANTAL FUNCTION, we ran 100 trials for $n = 25, 50, 75, 100, 150, 200$. We saw that the two methods returned the same solution for each $n$ for most trials. However, for each $n$, for 2 to 5 trials, R-SPLIT GREEDY outperformed INITIAL GREEDY and found solutions that were between 0.3% and 7% better.

### G.7 COMPUTER AND SOFTWARE INFRASTRUCTURE

We run all our experiments on Google Colab using libraries Pytorch, Numpy, and Itertools, which are available under licenses Caffe2, BSD, and CCA. Computer code for our algorithms and experiments is provided in https://anonymous.4open.science/r/Submodular-Set-Function-Optimization-8B0E/README.md.

## H DETAILS ON COMPUTING VOLUMES

The suprmodular rank $r$ functions on $\{0,1\}^n$ are a union of polyhedral cones in $\mathbb{R}^{2^{[n]}}$. We estimate their relative volume. We list the inequalities for each of the Minkowski sums of supermodular cones. We then sample 500,000 random points from a standard Gaussian, test how many of the points live in the union of the cones, and report the percentage. Table 1 shows these estimates. We see that the volume of the cones decreases with the number of variables $n$ and increases rapidly with the rank $r$. For instance, when $n = 4$ the volume of the set of submodular rank-2 functions, 5.9%, is nearly 1000 times larger than the volume of the set of submodular rank-1 functions, 0.0072%.

One may wonder if it is possible to obtain a closed form formula for these volumes. The relative volume, or solid angle, of a cone $C \subseteq \mathbb{R}^d$ is defined by

$$\mathrm{Vol}(C) := \int_{C \cap B} d\mu(x) / \int_B d\mu(x),$$

where $B = \{x \in \mathbb{R}^d \colon \|x\| \leq 1\}$ is the unit ball. In general there are no closed form formulas available for such integrals, even when $C$ is polyhedral. For simplicial cones there exist Taylor series expansions that, under suitable conditions, can be evaluated to a desired truncation level Ribando (2006). Instead of triangulating the cone of functions of bounded supermodular rank and then approximating the volumes of the simplicial components via a truncation of their Taylor series, we found it more reliable to approximate the solid angle by sampling. In our computations described in the first paragraph, we use

$$\mathrm{Vol}(C) = \int_C p(x) d\mu(x) \approx \frac{1}{N} \sum_{i=1}^{N} \chi_C(x_i),$$

where $C$ is the cone of interest (e.g., the cone of supermodular rank-$r$ functions), $\chi_C$ is the indicator function of the cone, $p$ is the probability density function of a zero centered isotropic Gaussian random variable and $x_i$, $i = 1, \ldots, N$ is a random sample thereof. To evaluate $\chi_C(x)$ we check if $x$ satisfies the facet-defining inequalities of any of the Minkowski sums that make up the cone, described in Theorem 12.

**Upper bound on the volume of the supermodular cone.** We use the correspondence in Proposition 27 to upper bound the volume of the supermodular cone on $n$ variables, as follows. There are $2^{n-1}$ distinct $\pi$-supermodular cones, all of the same volume and with disjoint interiors. Hence the volume of $\mathcal{L}_{(1,\ldots,1)}$ is upper bounded by $2^{-n+1}$, which decreases with linear rate as $n$ increases. We show that the volume decreases at least with quadratic rate.

**Proposition 78.** *The relative volume of $\mathcal{L}_{(1,\ldots,1)} \subseteq \mathbb{R}^{\{0,1\}^n}$ is bounded above by $0.85^{2^n}$. In particular, it decreases at least with quadratic rate as $n$ increases.*

*Proof.* We define $\mathcal{L}_0^{(n)} := \mathcal{L}_{(1,\ldots,1)}$ and denote the $i^{th}$ elementary supermodular cone on $n$ variables by $\mathcal{L}_i^{(n)}$. For distinct $i_1, \ldots, i_r \in [n]$, the cones $\mathcal{L}_0^{(n)}, \mathcal{L}_{i_1}^{(n)}, \ldots, \mathcal{L}_{i_r}^{(n)}$ have the same volume and disjoint interiors, and thus

$$(r + 1) \mathrm{Vol}(\mathcal{L}_0^{(n)}) \leq \mathrm{Vol}(\mathcal{L}_0^{(n)} + \mathcal{L}_{i_1}^{(n)} + \cdots + \mathcal{L}_{i_r}^{(n)}).$$

A function $f$ in $\mathcal{L}_0^{(n)} + \mathcal{L}_{i_1}^{(n)} + \cdots + \mathcal{L}_{i_r}^{(n)}$ consists of $2^r$ supermodular pieces on $n - r$ variables. That is, each pieces $f$ lies in $\mathcal{L}_0^{(n-r)}$. Thus $f$ lives in a Cartesian product of $2^r$ cones $\mathcal{L}_0^{(n-r)}$. Conversely, a function $f$ that lives in this product of cones lies in the Minkowski sum $\mathcal{L}_0^{(n)} + \mathcal{L}_{i_1}^{(n)} + \cdots + \mathcal{L}_{i_r}^{(n)}$. Hence

$$\mathrm{Vol}(\mathcal{L}_0^{(n)} + \mathcal{L}_{i_1}^{(n)} + \cdots + \mathcal{L}_{i_r}^{(n)}) = \mathrm{Vol}(\mathcal{L}_0^{(n-r)})^{2^r}.$$

Taking $r = n - 2$, and noting that $\mathcal{L}_0^{(2)} = \frac{1}{2}$, we obtain

$$\text{Vol}(\mathcal{L}_0^{(n)}) \leq \frac{2^{-2^{n-2}}}{n-1} \leq 0.85^{2^n}, \quad n \geq 2. \tag{6}$$

In particular, the relative volume of $\mathcal{L}_{(1,\ldots,1)} \subseteq \mathbb{R}^{2^{[n]}}$ decreases at least with quadratic rate. $\qquad\square$

Although equation 6 significantly improves the trivial upper bound, it is not clear how tight it is. For $n = 2$, the expression $\frac{2^{-2^{n-2}}}{n-1}$ in equation 6 equals 50%, which agrees with the true volume of $\mathcal{L}_0^{(2)}$. However, for $n = 3$ and $n = 4$ the values 12.5% and 2.1% it provides are larger than the experimentally obtained 3% and 0.0006% reported in Table 1. Nonetheless, the result shows that the relative volume of supermodular functions is tiny in high dimensions. This provides additional support and motivation to study relaxations of supermodularity such as our supermodular rank.

We are not aware of other works discussing the relative volume of supermodular cones. Following the above discussion, we may pose the following question:

**Problem 79.** *What is the relative volume of the cone of supermodular functions on $2^{[n]}$ in $\mathbb{R}^{2^{[n]}}$, and what is the asymptotic behavior of this relative volume as $n$ increases?*

## I   RELATIONS OF SUPERMODULAR RANK AND PROBABILITY MODELS

Probabilistic graphical models are defined by imposing conditional independence relations between the variables that are encoded by a graph, see Lauritzen (1996). Supermodularity (in)equalities arise naturally in probabilistic graphical models. We briefly describe these connections. We consider finite-valued random variables, denoted $X_i$, which take values denoted by lower case letters $x_i$.

**Conditional independence and modularity.**   Two random variables $X_1$ and $X_2$ are conditionally independent given a third variable $X_3$ if, for any fixed value $x_3$ that occurs with positive probability, the matrix of conditional joint probabilities $p(x_1, x_2|x_3) = p(x_1, x_2, x_3)/p(x_3)$ factorizes as a product of two vectors of conditional marginal probabilities,

$$p(x_1, x_2|x_3) = p(x_1|x_3)p(x_2|x_3),$$

for all $x_1$ in the range of values of $X_1$ and $x_2$ in the range of values of $X_2$. This means that the matrix of conditional joint probabilities has rank one, or, equivalently, that its $2 \times 2$ minors vanish. The vanishing of the $2 \times 2$ minors is the requirement that any submatrix obtained by looking at two rows and two columns has determinant zero,

$$p(x_1, x_2|x_3)p(x_1', x_2'|x_3) - p(x_1, x_2'|x_3)p(x_1', x_2|x_3) = 0,$$

for any two rows $x_1, x_1'$ and any two columns $x_2, x_2'$. Inserting the definition of conditional probabilities $p(x_1, x_2|x_3) = p(x_1, x_2, x_3)/p(x_3)$, moving the negative term to the right hand side, multiplying both sides by $p(x_3)p(x_3)$ and taking the logarithm, the rank one condition is rewritten in terms of log probabilities as

$$\log p(x_1, x_2, x_3) + \log p(x_1', x_2', x_3) = \log p(x_1, x_2', x_3) + \log p(x_1', x_2, x_3),$$

for all $x_1, x_1'$ in the range of values of $X_1$, all $x_2, x_2'$ in the range of values of $X_2$, and all $x_3$ in the range of values of $X_3$. Thus, with an appropriate partial order on the sample space, a conditional independence statement corresponds to modularity equations for log probabilities.

**Latent variables and supermodularity.**   If some of the random variables are *hidden* (or *latent*), characterizing the visible marginals in terms of (in)equality relations between visible margins becomes a challenging problem (see, e.g., Garcia et al., 2005; Allman et al., 2015; Zwiernik, 2015; Montúfar and Morton, 2015; Qi et al., 2016; Evans, 2018; Seigal and Montúfar, 2018). In several known cases, such descriptions involve conditional independence inequalities that correspond to submodularity or supermodularity inequalities of log probabilities.

Allman et al. (2015) studied $n$ discrete visible variables that are conditionally independent given a binary hidden variable. This is known as the 2-mixture of a $n$-variable independence model, and denoted by $\mathcal{M}_{n,2}$. The article Allman et al. (2015) shows that the visible marginals of $\mathcal{M}_{n,2}$ are characterized by the vanishing of certain $3 \times 3$ minors (these are equalities) and conditional independence inequality relations that impose that the log probabilities are $\pi$-supermodular for some partial order $\pi$. Thus, the set of log probabilities defined by inequalities of $\mathcal{M}_{n,2}$, discarding the equalities, gives a union of $\pi$-supermodular cones.

**Restricted Boltzmann machines and sums of supermodular cones.** A prominent graphical model is the restricted Boltzmann machine (RBM) (Smolensky, 1986; Hinton, 2002). The model $\mathrm{RBM}_{n,\ell}$ has $n$ visible and $\ell$ hidden variables, and defines the probability distributions of $n$ visible variables that are the Hadamard (entrywise) products of any $\ell$ probability distributions belonging to $\mathcal{M}_{n,2}$. Characterizing the visible marginals represented by this model has been a topic of interest, see for instance (Le Roux and Bengio, 2008; Montúfar et al., 2011; Martens et al., 2013). In particular, Cueto et al. (2010); Montúfar and Morton (2017) studied the dimension of this model, and Montúfar and Morton (2015); Montúfar and Rauh (2017) investigated certain inequalities satisfied by the visible marginals.

Seigal and Montúfar (2018) obtained a full description of the $\mathrm{RBM}_{3,2}$ model. In this case there are no equations, and the set of visible log probabilities is the union of Minkowski sums of pairs of $\pi$-supermodular cones. They proposed that one could study RBMs more generally in terms of inequalities and, to this end, proposed to study the Minkowski sums of $\pi$-supermodular cones, which remained an open problem in their work. We have provided a characterization of these sums in Theorem 12.

Proposition 45 implies that the model $\mathrm{RBM}_{n,\ell}$ is not a universal approximator whenever $\ell < \lceil \log_2 n \rceil + 1$. This does not give new non-trivial bounds for the minimum size of a universal approximator for $n \geq 4$, since the number of parameters of the model, $(n+1)(\ell+1) - 1$, is smaller than the dimension of the space, $2^n - 1$. However, the result shows that any probability distributions on $\{0,1\}^n$ whose logarithm is in the interior of the submodular cone requires at least $\ell = \lceil \log_2(n) \rceil$ to lie in $\mathrm{RBM}_{n,\ell}$. This complements previous results based on polyhedral sets called mode poset probability polytopes (Montúfar and Morton, 2015; Montúfar and Rauh, 2016).

