# OpenReview forum: "Supermodular Rank: Set Function Decomposition and Optimization"
_ICLR.cc/2024/Conference — Submitted to ICLR 2024_

### Official Review · Reviewer_HVke · 2023-10-29

**Soundness:** 3 good
**Presentation:** 1 poor
**Contribution:** 3 good
**Rating:** 5
**Confidence:** 3

**Summary:**

The paper studies the set function optimization problem, where there is a ground element set and the goal is to pick an element subset such that some certain objective is maximized. The authors mainly consider two concrete models: matroid-constrained maximization and set function ratio minimization. In the first model, we are given a monotone, non-negative function $f$ with generalized curvature $\alpha$ and submodularity ratio $\gamma$, and a matroid system $M$. The goal is to pick an element subset $S\in M$ such that $f(S)$ is maximized. The authors prove that there exists a framework such that by applying it to any approximation algorithm with $O(q(m))$ queries ($q(\cdot)$ is a polynomial function), a better approximation ratio can be obtained in time $O(2^{r-1}n^{r-1} q(n))$, where $r$ is the elementary submodular rank. In the second model, we are given two set functions $f, g$ and the goal is to pick a subset $S$ such that $f(S)/g(S)$ is minimized. The authors prove that when $f,g$ are normalized positive monotone functions and $f$ has a bounded elementary submodular rank, they obtain an approximation ratio polynomially that was previously only available when function $f$ was submodular. Finally, the authors conduct experiments to investigate the empirical performance of the algorithms.

**Strengths:**

- The paper considers two classical and important models in set function optimization. The main contribution of the paper is introducing the concept of elementary submodular rank and building on it to extend the previous results to a more general function class.

- Both theoretical analyses and experimental evaluations for the proposed algorithms are provided in the paper.

**Weaknesses:**

- A main weakness is that the paper is not well-written. The structure is quite confusing. The formal definitions of the considered models are not provided until section 4. Several new definitions are introduced before Section 4. However, the absence of any intuitive explanations renders the paper less accessible to readers.

**Questions:**

(1) Could you give some intuition about the elementary-submodular-rank-based trick used in the paper?

---

> ### Author Response · Authors · 2023-11-21
>
> > A main weakness is that the paper is not well-written. The structure is quite confusing. The formal definitions of the considered models are not provided until section 4. Several new definitions are introduced before Section 4. However, the absence of any intuitive explanations renders the paper less accessible to readers.
>
> We thank the reviewers for their comments. We have updated the manuscript to add more explanatory examples to improve the paper’s readability. The changes are highlighted in blue.
>
> > (1) Could you give some intuition about the elementary-submodular-rank-based trick used in the paper?
>
> The intuition might be more apparent if we think of continuous functions. The idea behind the elementary rank can be thought of (not strictly true, but ok for intuition reasons) as determining the number of coordinate directions in which the function is non-convex. Once we have determined these directions, we decompose our function into convex orthogonal functions. Then, we independently optimize these functions.

---

> > ### Comment · Reviewer_HVke · 2023-11-22
> >
> > The reviewer thanks the author for their response and the newly added examples in the revision. I believe this is an interesting work. However, the paper seems to still need a cleaner overall structure to make the main contributions more readable.

---

> > > ### Author Response · Authors · 2023-11-22
> > > **thank you**
> > >
> > > We sincerely thank the reviewer for their comment. Our paper is currently structured as follows.
> > >
> > > 1. Section 2 provides the mathematical background on supermodular functions and cones
> > > 2. Section 3 defines the supermodular rank
> > > 3. Section 4 provides an application of supermodular rank to set function optimization.
> > > 4. Section 5 presents numerical experiments.
> > >
> > > We kindly ask what structural changes the reviewer would like to see.

---

> > > > ### Comment · Reviewer_HVke · 2023-11-22
> > > >
> > > > The reviewer thanks the author for their further response. For me, it's more natural to introduce the target optimization problems (Section 4) as soon as possible, so that the reader can get the main goal quickly. And then state why defining the supermodular rank is helpful and necessary. This seems a better way to organize the paper, but this is just my personal opinion, the authors should avoid overfitting.

---

### Official Review · Reviewer_jvwD · 2023-10-30

**Soundness:** 3 good
**Presentation:** 1 poor
**Contribution:** 3 good
**Rating:** 6
**Confidence:** 3

**Summary:**

This paper considers the supermodular and submodular optimization problem. In these problems, we are given a supermodular/submodular or a related function defined over a ground set. The goal is to select a certain subset of the ground elements such that (1) the selected subset satisfies some properties; (2) the value of the selected subset is optimized. If a function is not submodular/supermodular, one can describe it into several parameters, and the approximation ratio shall also be related to these parameters.

The main contribution of this work is a new approach to grading the space of set function. They propose a new concept called supermodular/submodular rank, which is defined over a partial order set. Based on such a concept, they show that a function can be decomposed into a summation of several \p-supermodular/submodular functions. Then, one can improve the approximation by such a splitting.

**Strengths:**

1. The high-level idea of this paper is clear. The main technical idea is to decompose a function into a sum of functions that are \pi-supermodular/submodular. And then split the problem into several submodular pieces. This improves the approximation when the submodular/supermodular rank is bounded.

2. This paper is technically involved. To my knowledge, there is no such definition and decomposition in the literature. Probably the most related one is that a submodular function can be decomposed into n! additive functions, but the definition used in this paper is quite different from this.

**Weaknesses:**

1. The presentation of this work is poor. It seems that the authors ran out of space and moved a lot of background knowledge to the appendix. Without this knowledge, it's hard to get the definition of \pi-supermodular. After moving, it seems that the authors didn't do careful proofreading. For example, R(alpha, gamma) in Theorem 25 is not defined. This significantly impairs readability. I appreciate that the authors also try to explain their ideas with some examples, and I also understand that the space issue is not the authors' fault, but it is a fact that the paper does need a better presentation.

2. To my understanding of Table 2, the proposed algorithm improves the previous ratio only in the case where the elementary submodular rank is a constant. If this is true, it’s not clear how important this improvement is. Because the paper didn't include a discussion about whether there exists some famous functions with a constant submodular rank. Besides this, the paper only includes an upper bound on the rank of a function but excludes the way to compute the rank of a function. Maybe I missed something, and such a computation is trivial, but this should be stated explicitly.

**Questions:**

See my second comment in Weaknesses.

---

> ### Author Response · Authors · 2023-11-21
>
> > The presentation of this work is poor. It seems that the authors ran out of space and moved a lot of background knowledge to the appendix. Without this knowledge, it's hard to get the definition of \pi-supermodular. After moving, it seems that the authors didn't do careful proofreading. For example, R(alpha, gamma) in Theorem 25 is not defined. This significantly impairs readability. I appreciate that the authors also try to explain their ideas with some examples, and I also understand that the space issue is not the authors' fault, but it is a fact that the paper does need a better presentation.
>
> We thank the reviewers for their comments. We have updated the manuscript to add more explanatory examples to improve the paper’s readability. The changes are highlighted in blue.
>
> > To my understanding of Table 2, the proposed algorithm improves the previous ratio only in the case where the elementary submodular rank is a constant. If this is true, it’s not clear how important this improvement is. Because the paper didn't include a discussion about whether there exists some famous functions with a constant submodular rank. Besides this, the paper only includes an upper bound on the rank of a function but excludes the way to compute the rank of a function. Maybe I missed something, and such a computation is trivial, but this should be stated explicitly.
>
> Computing the rank of a function (unless we know something about the function) is not easy. We do not have a simple method for doing so. However, we can provide examples of functions that have low submodular rank.
>
> **RBM**
>
> The first example comes from Restricted Boltzmann Machines (RBMs). These are graphical models for modeling probability distributions on $\{0,1\}^n$. They have a hyperparameter $m$, the number of hidden nodes. Prior work (See references [Allman et al., 2015]) has shown that when $m=1$, this model can represent distributions $f(x)$ if and only if $\log f$ is $\pi$-supermodular and satisfies certain polynomial equality constraints.
> Then, when we move to more significant values of $m$, it can be shown that the model can model distribution $f(x)$ only if $\log f$ is rank-$m$ supermodular. Thus log RBM distributions are functions with bounded supermodular rank. In this case, the optimization problem would correspond to finding modes of the distribution.
>
> **One hidden layer neural networks**
>
> Building on this, we have the following.
>
> **Proposition:** Let $f$ be a real-valued function on $\{0,1\}^n$  that is the composition of an affine function and a convex function. That is, $f(x) = \phi(wx + c)$ where $w$ is a vector of length $1 × n$ and $c$ is a scalar and $\phi : \mathbb{R} \to \mathbb{R}$  is convex. Then $f$ is sign(w)-supermodular.
>
> *Proof*:We show that the elementary imset inequalities are satisfied.
> Such an inequality involves four vectors on a two-dimensional face of the cube $\{0, 1\}^n$. We have two indices $x_i,x_j$ that vary and a fixed value of $x_{[n]\setminus\{i,j\}}$, the vector $x$ restricted to the set $[n]\setminus\{i,j\}$.
> Let $y$ be the entry of the face where $x_i = x_j = 0$.
> Letting $c' = c + Ay$, we seek to compare $\phi(c' + w_i) + \phi(c' + w_j)$ with $\phi(c') + \phi(c' + w_i + w_j)$.
> By the definition of ${\rm sign}(w)$-supermodularity,
> if both entries $w_i$ and $w_j$ have the same sign, we require
> $$\phi(c' + w_i) + \phi(c' + w_j) \leq \phi(c') + \phi(c' + w_i + w_j) $$
> while if $w_i w_j \leq 0$, we require
> $$\phi(c' + w_i) + \phi(c' + w_j) \geq \phi(c') + \phi(c' + w_i + w_j) .$$
> The inequalities hold by the fact that $\phi$ is convex.
> For example, if $w_i,w_j\geq0$, then $c\leq c+w_i, c+w_j\leq c+w_i+w_j$, and we apply the definition of convexity as applied to a comparison of four points.
>
> **Thus, in particular, a one-hidden layer ReLU neural network, when restricted to $\\{0,1\\}^n$ with $k$ hidden nodes with positive outer weights, is rank-$k$ supermodular.**

---

> > ### Comment · Reviewer_jvwD · 2023-11-23
> >
> > I'd like to thank the authors' response. I am satisfied with my second question. It is particularly interesting to see that there is a rank-k-supermodular function in neural networks. I believe this is an interesting work, and it's a good paper, but not in its current version due to the presentation.

---

### Official Review · Reviewer_Pie2 · 2023-10-30

**Soundness:** 3 good
**Presentation:** 1 poor
**Contribution:** 1 poor
**Rating:** 3
**Confidence:** 5

**Summary:**

This work measures how far a function $F$ from being submodular or supermodular. The main idea is the decomposition of $F$ into the sum of the smallest number $r$ of submodular function with a different total order on individual variables. The number $r$ is the submodular rank of $F$. The less interesting part of this work is the elementary submodular rank where the order can be reversed in one variable at most for each function. The paper proposes a simple R-SPLIT algorithm using the proposed notion, which splits up a function $f$ of rank $r+1$ into $2^r$ pieces and runs a simple algorithm e.g. greedy on each piece returning the best solution.

**Strengths:**

It is clear that this work is novel in terms of the definitions of supermodular/submodular rank and elementary rank. This work also provides both theoretical results and empirical evaluation.

**Weaknesses:**

The theoretical contribution of this work seems to be very weak. In particular, the elementary rank $r$ basically says that the function becomes submodular for all assignments to a subset of the variables, which leads to an exhaustive search for this subset. Hence, demonstrated by the complexity of the algorithmic part, the contribution of this work does not meet the bar of top-tier conferences such as ICLR. I have two additional comments:

1. The current paper is very tough to read.
2. The empirical evaluation is not convincing.

My suggestion to the authors is to consider submitting this paper to other venues such as ICALP, SODA, and ESA. The main reason behind this suggestion is that I think the paper can be presented much better without the practical part (which is not convincing in my opinion), which can be replaced by highlighting some non-trivial theoretical results such as Theorem 10.

**Questions:**

1) Seems like when the submodular rank $r$ is high, the algorithms are impractical.
2) Empirical evaluation is not convincing.

---

> ### Author Response · Authors · 2023-11-21
>
> > The theoretical contribution of this work seems to be very weak. In particular, the elementary rank r basically says that the function becomes submodular for all assignments to a subset of the variables, which leads to an exhaustive search for this subset.
>
> As the reviewer pointed out, the notion of supermodular rank is novel and interesting. Please see the general response for an explanation of our contributions.
>
> The elementary supermodular rank is a particular case where we restricted the types of permutations that were allowed. Here, apriori, there is no reason that this notion would simplify to such an elementary property, but the fact that it does is quite interesting.
>
> Building on this, since the idea of supermodular rank is novel, this gradation of the space of set functions is new. It provides a valuable way to think about the complexity of set function optimization.
>
> Regarding the complexity of our algorithms, recall that there is no free lunch in optimization. Our paper provides a lower bound that says that set function optimization for a rank $r$ function necessarily requires $2^r$ function evaluations. Our algorithms can maintain a low complexity for optimizing functions with low elementary supermodular rank. By necessity, the complexity of optimizing general objective functions, or functions that have a high elementary supermodular rank, is high. To illustrate this more concretely, we can provide examples that serve as a lower bound as follows.
>
> Let $\hat{f}$ be your favorite submodular function on a set of size $n-r$, and define $f$  to be an elementary rank-$r$ submodular function with pieces given by $\hat{f}+c_k$. All of the pieces of $f$ are just shifts of $\hat{f}$. To get an $O(1)$ approximation algorithm for general $c_k$'s, at the very minimum, we need to evaluate the function $f$ once at each of the $2^r$ shifts. Thus, we will always need at least $\Omega(2^r)$  time. As mentioned in Remark 27, if the decomposition is known, which is the case for the lower bound, the only exponential term in the time complexity for R-split is $2^r$. Thus, the algorithm achieves the complexity lower bound.
>
> We would also like to highlight that our algorithm is different from other methods. In particular, typically, algorithms for submodular optimization, such as the plain greedy, are analyzed for things like weak submodularity, but no changes are made to the algorithm to deal with the lack of submodularity. We exploit the non-submodularity structure by providing gradation on the space of functions.
>
> In light of the above strengths, the fact our algorithm is additionally *simple* is an additional strength.
>
> > Hence, demonstrated by the complexity of the algorithmic part, the contribution of this work does not meet the bar of top-tier conferences such as ICLR.
>
> The primary contribution of the paper is not algorithmic but theoretical. The main contributions are
> 1. The definition of rank
> 2. The proof of the maximal rank
> 3. The gradation of the space set functions
> 4. The results show that theoretical guarantees obtained for submodular functions can be lifted to higher ranks but with a necessary penalty. That is, this penalty cannot be avoided in some cases.
>
> > The current paper is very tough to read.
>
> We thank the reviewers for their comments. We have updated the manuscript to add more explanatory examples to improve the paper’s readability. The changes are highlighted in blue.
>
> > The empirical evaluation is not convincing.
>
> We kindly disagree. The experimental results show a consistent and significant improvement in the performance compared to Greedy. For example,
>
> 1. Figure 1a shows that the theoretical bound improves by 400\% in one instance.
> 2. Figure 1b shows orders of magnitude decrease in the error
> 3. Figure 2a shows a consistent improvement as well.
> 4. Figure 2b shows that in most cases, going from greedy to 1-split greedy, the percentage of times we find the optimal set more than doubles.
> 5. Figure 2c shows that it scales to large problems, and we consistently have over 5\% improvement.
>
> > Seems like when the submodular rank $r$  is high, the algorithms are impractical.
>
> Yes, but we provide an impossibility result in approximating the solution to a related problem with less work.

---

> > ### Comment · Reviewer_Pie2 · 2023-11-21
> >
> > I would like to thank the authors for their response. My initial understanding of the paper would appear to be correct.

---

> > > ### Author Response · Authors · 2023-11-22
> > >
> > > We thank the reviewer for their comment, but we kindly disagree with the reviewer. The experimental evidence does show significant improvements and we have updated the paper (available on openreview) to improve readability.

---

### Official Review · Reviewer_1L31 · 2023-11-10

**Soundness:** 3 good
**Presentation:** 2 fair
**Contribution:** 3 good
**Rating:** 6
**Confidence:** 2

**Summary:**

The authors introduce the concept of supermodular rank for functions defined on partially ordered sets. Supermodular rank characterizes how a function can be decomposed into a sum of functions that exhibit a "supermodular" property. This concept allows for a more refined understanding of the structure of set functions. The authors propose optimization algorithms, namely R-SPLIT and R-SPLIT RATIO, for optimizing monotone set functions and the ratio of set functions. These algorithms provide a trade-off between computational cost and accuracy, offering theoretical guarantees for their performance.

**Strengths:**

1. Introduction of Supermodular Rank: The concept of supermodular rank is considered interesting and valuable for understanding the structure of set functions.

2. Optimization Algorithms: The proposed optimization algorithms, R-SPLIT and R-SPLIT RATIO, are seen as valuable contributions. They provide a trade-off between computational cost and accuracy while offering theoretical guarantees for their performance.

**Weaknesses:**

1. Complex and Notation-Heavy: The paper is noted as being quite complex and filled with notation, making it challenging to follow. Simplifying the presentation or providing additional explanations could enhance the accessibility of the material.

2. Lack of Clarity on Performance Improvement: The paper's comparison to existing solutions, particularly in Table 2, is mentioned as lacking clarity. It is not immediately clear how the proposed algorithm outperforms existing solutions. The authors should elaborate on the additional benefit brought by the proposed algorithm's computational overhead.

**Questions:**

See Weakness 2 above.

---

> ### Author Response · Authors · 2023-11-21
> **Response to Reviewer 1L31**
>
> We thank the reviewer for their feedback.
>
> ​> Complex and Notation-Heavy The paper is noted as being quite complex and filled with notation, making it challenging to follow. Simplifying the presentation or providing additional explanations could enhance the accessibility of the material.
>
> We thank the reviewers for their comments. We have updated the manuscript to add more explanatory examples to improve the paper’s readability. The changes are highlighted in blue.
>
> > Lack of Clarity on Performance Improvement: The paper's comparison to existing solutions, particularly in Table 2, is mentioned as lacking clarity. It is not immediately clear how the proposed algorithm outperforms existing solutions. The authors should elaborate on the additional benefit brought by the proposed algorithm's computational overhead.
>
> The table should be read not as saying that we improve the performance of the method (which we do show happens empirically), but in a different manner. Specifically, prior work showed that for a family of functions $\mathcal{F}$, we can **prove** that we have a good approximation rate. With our framework, we can show that this approximation rate can be lifted to a larger family of functions $\mathcal{G}$ such that $\mathcal{F} \subset \mathcal{G}$. Table 1 shows that the size of $\mathcal{G}$ can be significantly bigger than $\mathcal{F}$.

---

### Author Response · Authors · 2023-11-21
**General Response**

We thank the reviewers for their comments. We have updated the manuscript to add more explanatory examples to improve the paper’s readability.

Additionally, we would like to reiterate the contributions of our paper.

**Theoretical Contributions**

Set function optimization is an important problem that has many applications. Prior work has shown that if the function has a favorable structure, particularly if it is submodular, then it is easy to optimize. However, on the other extreme, if we have set functions that have no structure, then it is clearly impossible to do better than to evaluate the function at all $2^n$ possible inputs.

However, as $n$ grows, the percentage of submodular set functions decreases exponentially, as shown in our paper. Prior work has tried to define a broader class of functions with favorable structure by introducing quantities such as submodularity ratio and curvature. These works then analyze the performance of Greedy on this broader class of functions and show that the performance degrades and hence we have bad approximations.

However, we take a complementary approach. We take our set of functions and partition the whole space by defining a measure of complexity on the space. **Namely, submodular rank.** We then show that the maximum rank is $O(\log(n))$. Using this, we show that as the rank $r$ increases from 1 to $O(\log(n))$, the complexity of getting good approximations of the optimizer grows exponentially in $r$. We then complement this result by showing that our imposed structure presents a way to optimize these functions. **Unlike prior work, the idea is to present methods that do not degrade the approximation quality.**

**Empirical Contributions**

We show that for small values of $r$, our method is feasible for large problems. Furthermore, we show that our method provides consistent improvement.

**Functions with low supermodular rank**

The first example comes from Restricted Boltzmann Machines (RBMs). These are graphical models for modeling probability distributions on $\{0,1\}^n$. They have a hyperparameter $m$, the number of hidden nodes. Prior work (See references [Allman et al., 2015]) has shown that when $m=1$, this model can represent distributions $f(x)$ if and only if $\log f$ is $\pi$-supermodular and satisfies certain polynomial equality constraints.
Then, when we move to more significant values of $m$, it can be shown that the model can model distribution $f(x)$ only if $\log f$ is rank-$m$ supermodular. Thus log RBM distributions are functions with bounded supermodular rank. In this case, the optimization problem would correspond to finding modes of the distribution.

**One hidden layer neural networks**

Building on this, it is easy to see the following.

Let $f$ be a real-valued function on $\{0,1\}^n$ that is the composition of an affine function and a convex function. That is, $f(x) = \phi(wx + c)$ where $w$ is a vector of length $1 × n$ and $c$ is a scalar and $\phi : \mathbb{R} \to \mathbb{R}$  is convex. Then $f$ is sign(w)-supermodular.

Thus, in particular, a one-hidden layer ReLU neural network, when restricted to $\{0,1\}^n$ with $k$ hidden nodes with positive outer weights, is rank-$k$ supermodular.

---

### Meta-Review · Area_Chair_Yp41 · 2023-12-06

**Metareview:**

The paper introduces the concept of supermodular rank for a function on a lattice, which is the smallest number of functions in a decomposition of the function into supermodular functions. The paper designs algorithms that are able to leverage a low supermodular rank structure in order to maximize such functions subject to constraints.

The reviewers appreciated the novelty of the definition of supermodular rank. The reviewers remained concerned about the significance of the main contributions, the practical applicability of this work, and the presentation. The overall consensus was that the paper does not meet the threshold for acceptance.

**Justification For Why Not Higher Score:**

The theoretical results of the paper seem to be weak, and the proposed algorithms have limited practical applicability due to high running times.

**Justification For Why Not Lower Score:**

N/A

---

### Decision · Program_Chairs · 2024-01-16

Reject